# Conditional Variable Flow Matching: Transforming Conditional Densities with Amortized Conditional Optimal Transport

## Abstract

Forecasting stochastic nonlinear dynamical systems under the influence of conditioning variables is a fundamental challenge repeatedly encountered across the biological and physical sciences. While flow-based models can impressively predict the temporal evolution of probability distributions representing possible outcomes of a specific process, existing frameworks cannot satisfactorily account for the impact of conditioning variables on these dynamics. Amongst several limitations, existing methods require training data with paired conditions and are developed for discrete conditioning variables. We propose *Conditional Variable Flow Matching* (CVFM), a framework for learning flows transforming conditional distributions with amortization across continuous conditioning variables – permitting predictions across the conditional density manifold. This is accomplished through several novel advances. In particular, simultaneous sample conditioned flows over the main and conditioning variables. In addition, motivated by theoretical analysis, a conditional Wasserstein distance combined with a loss reweighting kernel facilitating conditional optimal transport. Collectively, these advances allow for learning system dynamics provided measurement data whose states and conditioning variables are not in correspondence. We demonstrate CVFM on a suite of increasingly challenging problems, including discrete and continuous conditional mapping benchmarks, image-to-image domain transfer, and modeling the temporal evolution of materials internal structure during manufacturing processes. We observe that CVFM results in improved performance and convergence characteristics over alternative conditional variants.

## 1 Introduction

Appropriately modeling the time-dependent evolution of distributions is a central goal in multiple scientific fields, such as single-cell genomics (Tong et al., 2023a; Bunne et al., 2023a), meteorology (Fisher et al., 2009), robotics (Ruiz-Balet & Zuazua, 2023; Chen et al., 2021), and materials science (Kalidindi, 2015; Adams et al., 2013). In each of these fields, forecasting stochastic nonlinear dynamical systems requires a methodology for learning the transformations of time-evolving densities given *unpaired* observational samples, or observations across time which are not in correspondence. This requirement arises due to practical constraints on data collection in these scientific applications. For example, in both single-cell genomics and materials science, experimental testing to quantify the system's state is often destructive, precluding measurement of the state across multiple time steps (Tong et al., 2023a; Bunne et al., 2023a; Ghanavati & Naffakh-Moosavy, 2021; ASTM International, 2024).

Various approaches to address this challenge have recently been proposed, including diffusion Schrödinger bridges (DSB) (Liu et al., 2022a; Chen et al., 2023; De Bortoli et al., 2021; Bunne et al., 2023a; Tang et al., 2024) alongside extensions of Flow Matching (FM) (Tong et al., 2023b;a). These approaches generalize denoising diffusion probabilistic models (Ho et al., 2020), score matching (Song et al., 2021), and FM (Lipman et al., 2023; Albergo & Vanden-Eijnden, 2023; Liu et al., 2022b), to arbitrary source distributions – a necessary relaxation to model the evolutionary

pathways of physical or biological systems, as such natural systems rarely exhibit Gaussian source distributions[1].

Despite the apparent success of such approaches, their practical utility has been limited, solely permitting the simulation of evolving unconditional distributions. In modeling the dynamics of real systems, the most paramount questions are often variants of, *how might an intervention affect the resulting dynamics*? Addressing such questions requires the ability to uncover the behavior of *conditional* stochastic dynamical systems, despite our limited capacity to inspect their time-dependent states. In addition to *unpaired* state measurements, we often must also contend with unpaired conditioning. Apart from the destructive nature of common data collection in the sciences, unpaired conditioning commonly arises due to the prohibitive costs sample acquisition. Design-of-experiments or active learning approaches are frequently employed to mitigate these costs by minimizing the number of experiments – identifying a series of maximally informative test configurations (Lookman et al., 2019; Tran et al., 2020). Although, this diversity ensures that conditioning is purposefully rarely repeated.

Extensions modeling the dynamics of conditional distributions are still in their infancy (Ho & Salimans, 2022; Zheng et al., 2023; Bunne et al., 2022; Harsanyi et al., 2024; Bunne et al., 2023b; Isobe et al., 2024). Conditional input convex neural networks (ICNN) (Bunne et al., 2022; Harsanyi et al., 2024; Bunne et al., 2023b) and conditional extensions of flow matching (Isobe et al., 2024; Zheng et al., 2023; Dao et al., 2023), in particular, require datasets with matching conditioning, or the ability to first be able to select $y \in \mathcal{Y}$ and subsequently sample $x_t \sim p_t(x|y)$. This structure degrades in the limit of continuous conditioning variables, where obtaining multimarginal samples with equivalent conditioning is infeasible; a setting frequently encountered in engineering applications, such as in the monitoring and modeling of manufacturing processes. In this application of stochastic dynamics in materials science, the resulting material structure depends upon process parameters such as applied temperature or power (Liu et al., 2022c; Schrader & Elshennawy, 2000). Select recent works have explored extensions to enable this continuous conditioning treatment by introducing approximate conditional Wasserstein distances which more severely penalize transport across the conditioning variable (Chemseddine et al., 2024; Kerrigan et al., 2024). Detrimentally, we demonstrate in our work that this approximation in isolation is both insufficient to adequately capture conditional dynamics of for complex mappings and is practically challenging to calibrate due to highly sensitive, problem specific hyperparameter selection.

**We propose *Conditional Variable Flow Matching* (CVFM), a general approach for learning the flow between source and target conditional distributions**. Importantly, CVFM supports *entirely unpaired datasets*, wherein neither the sample data nor their corresponding conditioning variables need to be paired. We motivate CVFM's proposed training algorithm through a theoretical analysis of the stability of flow matching on conditional densities. Here, we observe the need for optimal transport over the conditioning variables. To realize this, core to CVFM is the usage of two conditional flows, a conditional Wasserstein distance, and a condition dependent loss reweighting kernel, generalizing existing simulation-free objectives for continuous normalizing flows (CNF) (Lipman et al., 2023; Albergo & Vanden-Eijnden, 2023) and stochastic dynamics (Tong et al., 2023a; Shi et al., 2023) to the conditional setting. We specifically focus on dynamics wherein the marginal distribution over the conditioning variable remains constant – a common setting in applied problems. The algorithm is introduced in the dynamical formulation of optimal transport (OT), augmenting FM and ordinary differential equation (ODE) based transport in defining a straightforward training objective for learning amortized conditional vector fields. Our objective facilitates simulation across the conditional density manifold, leveraging a conditional Wasserstein distance and kernel for enabling conditional OT. Subsequently, we analyze CVFM on several toy problems, demonstrating its superior performance and convergence behavior compared to existing methods. The performance of CVFM is further demonstrated in two more challenging case studies: the dynamics of material microstructure evolution conditioned on various manufacturing processes, and class conditional image-to-image mapping. As a part of these latter case studies, we further demonstrate the applicability of our method to approximating conditional Shrödinger bridges with a score-based stochastic differential equation (SDE) extension to FM (Tong et al., 2023a).

---

[1]We note that when approximating the dynamics of real systems, the model is trained to transform the state density between successive time steps. Therefore, even if the density at $t = 0$ is Gaussian, the bridge between arbitrary time steps must be generalized.

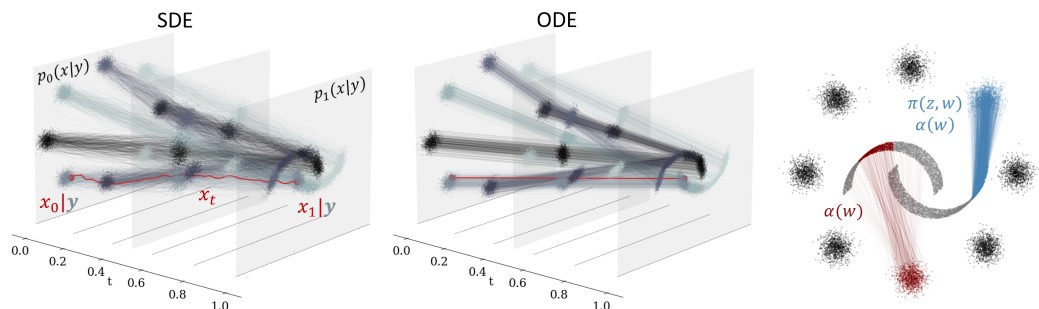

Figure 1. (Left) Conditional time-dependent density evolution from 8-Gaussians to Moons through the SDE and ODE formulations of CVFM. (Right) Comparison of conditional flows learned using just the proposed conditional kernel (**red**) and the proposed CVFM framework (**blue**). The kernel effectively further facilitates disentanglement of the flow conditioning during static conditional OT.

## 2 DYNAMIC MASS TRANSPORT METHODS

### 2.1 FLOW MATCHING

Continuous Normalizing Flows (CNF) (Chen et al., 2018) define a mapping between distributions $p(x_0)$ and $p(x_1)$ both on the same domain, $x_0, x_1 \in \mathbb{R}^N$ via the following ordinary differential equation (ODE).

$$\frac{d}{dt}\phi_t(x) = u_t(\phi_t(x)), \quad \phi_0(x) = x \tag{1}$$

This ODE defines a flow, $\phi_t(x) = \phi(x, t)$, producing a push-forward operation for transforming an initial distribution, $p_0(x)$, into an arbitrary time dependent distribution, $p_t(x)$, (i.e., $\phi_\# : [0,1] \times \mathcal{P}(\mathbb{R}^N) \to; \mathcal{P}(\mathbb{R}^N)$), such that $p_t(x)$ is equal to $p(x_1)$ at $t = 1$ (Lipman et al., 2023; Albergo & Vanden-Eijnden, 2023; Liu et al., 2022b). Individual samples $x_0 \sim p(x_0)$ can be transformed to $x_1 \sim p(x_1)$ by integrating the vector field $u_t : [0,1] \times \mathbb{R}^N \to \mathbb{R}^N$ and solving the ODE in Eq. (1).

Flow matching (FM) provides a simulation-free objective for constructing the *marginal probability path* $p_t(x)$ via a marginalization of sample conditioned probability paths $p_t(x|z)$, conditioned on observations $z = (x_0, x_1)$ drawn from the empirical distributions $q(x_0)$ and $q(x_1)$.

$$p_t(x) = \int p_t(x|z)q(z)dz \tag{2}$$

Lipman et al. (2023) demonstrates that one can similarly marginalize over conditional vector fields $u_t(x|z)$, whose marginal $u_t(x)$ generates the probability flow $p_t(x)$ (Theorem 1 (Lipman et al., 2023)). The consequences of which permit directly regressing upon the conditional vector field as

$$\mathcal{L}_{\text{CFM}}(\theta) = \mathbb{E}_{t, q(z), p_t(x|z)} ||v_\theta(x, t) - u_t(x|z)||^2 \tag{3}$$

where for conditional Gaussian paths $p_t(x|z) = \mathcal{N}(x|\mu_t(z), \sigma_t^2(z))$ of $\phi_{t,z}(x) = \mu_t(z) + \sigma_t(z)x$ the unique conditional vector field $u_t(x|z)$ can be solved in closed form (Theorem 3 (Lipman et al., 2023), Theorem 2.1, (Albergo & Vanden-Eijnden, 2023)).

### 2.2 OPTIMAL TRANSPORT

The optimal transport (OT) problem aims to identify a mapping between measures, $\nu$ and $\mu$, with minimal displacement cost (Villani, 2009). The Kantorovich relaxation attempts to recover the OT coupling $\pi$ given the potential set of all couplings on $\mathbb{R}^N \times \mathbb{R}^N$, such that the coupling's marginals are the original distributions, $\Pi(\mu, \nu) = \{\pi \in \mathcal{P}(\mathcal{X} \times \mathcal{Y}) : P_{\mathcal{X}\#}\pi = \mu \text{ and } P_{\mathcal{Y}\#}\pi = \nu\}$. The resulting distance $W(\mu, \nu)$ is the Wasserstein distance

$$W(\mu, \nu) = \inf_{\pi \in \Pi(\mu, \nu)} \int c(x, y) \pi(x, y) dx dy \tag{4}$$

where the 2-Wasserstein distance $W(\mu, \nu)_2^2$ is induced by the ground cost $c(x, y)^2$.

**Entropically-Regularized OT**: The optimization of the Wasserstein distance over couplings $\Pi(\mu, \nu)$ is a computationally challenging problem. The seminal work by Cuturi (2013) alleviates these issues by introducing a regularization term using the Shannon entropy $H(\pi)$, equivalently described by the KL-divergence $\mathrm{KL}(\pi \| \mu \otimes \nu)$ between a coupling, $\pi$ and independent joint distribution between $\mu$ and $\nu$, $\mu \otimes \nu$ (Khan & Zhang, 2022).

$$W_\varepsilon(\mu, \nu) = \inf_{\pi \in \Pi(\mu, \nu)} \int c(x, y) \pi(x, y) dx dy - \varepsilon \mathrm{KL}(\pi \| \mu \otimes \nu) \tag{5}$$

As $\varepsilon \to 0$ we recover the Kantorovich optimal transport plan, which we will distinguish as *exact optimal transport*, while $\varepsilon > 0$ yields a differentiable approximation to Eq. (4) with respect to the inputs.

**OT-FM**: While the Gaussian conditional probability paths for $p_t(x|z)$ in FM are the OT paths after conditioning on $z$ (Peyré & Cuturi, 2020), the induced marginal flow, defining $p_t(x)$, does not provide OT between distributions. Recent works have demonstrated that *dynamic* marginal OT can be achieved through identifying the *static* OT map (Tong et al., 2023b; Pooladian et al., 2023). Practically, this is achieved by sampling $z$ according to the distribution $q(z) = \pi^*(x_0, x_1)$, where $\pi^*$ denotes the OT coupling. In practical implementations, $\pi^*$ is identified within minibatches during training. A core benefit of this approach is a pronounced reduction in variance of the regression target in Eq. (3), enabling expedited model convergence.

## 2.3 SCHRÖDINGER BRIDGE

The Schrödinger bridge problem aims to identify the most likely stochastic mapping between arbitrary marginal distributions $\mathbb{P}_0 = \mu_0$ and $\mathbb{P}_1 = \mu_1$ with respect to a given reference process $\mathbb{Q}$ (Cuturi, 2013; Léonard, 2013; Schrödinger, 1932), defined as

$$\mathbb{P}_t^* = \operatorname*{argmin}_{\mathbb{P}_0 = \mu_0, \mathbb{P}_1 = \mu_1} \mathrm{KL}(\mathbb{P}_t \| \mathbb{Q}_t) = \int_{\mathcal{C}[0,1]} \log\left(\frac{d\mathbb{P}_t}{d\mathbb{Q}_t}\right) d\mathbb{P}_t \tag{6}$$

where $\mathcal{C}[0, 1]$ denotes continuous paths over $\mathbb{R}^N$ over the time interval $[0, 1]$ and $\frac{d\mathbb{P}_t}{d\mathbb{Q}_t}$ the Raydon-Nikodym derivative. Frequently, $\mathbb{Q}$ is taken to be $\mathbb{Q} = \sigma \mathbb{W}$, where $\mathbb{W}$ is standard Brownian motion, otherwise known as the *diffusion Schrödinger bridge* (De Bortoli et al., 2021; Chen et al., 2022; Vargas, 2021; Bunne et al., 2023a; Shi et al., 2023).

Prior work has elucidated a rich relationship between the Schrödinger bridge problem and OT, in particular entropy-regularized OT (Léonard, 2013; Mikami & Thieullen, 2006; Mikami, 2004; Léonard, 2010). More specifically, with the reference process assumed as standard Brownian motion, the marginals of the dynamic Schrödinger bridge can be considered to be a mixture of Brownian bridges weighted by the *static* entropic OT map (Léonard, 2010; 2013)

$$p_t(x) = \int p_t(x|x_0, x_1) d\pi_\varepsilon^*(x_0, x_1). \tag{7}$$

In this reframing, the diffusion Schrödinger bridge can be approximated through a collection of marginal Brownian bridges defined as $p_t(x|x_0, x_1) = \mathcal{N}(x; (1 - t)x_0 + tx_1, \sigma^2 t(1 - t))$, with diffusion coefficient $\sigma$; a construction reminiscent of Eq. (2).

## 2.4 SCORE AND FLOW MATCHING

Given the notable connection between the Shrödinger bridge problem (Schrödinger, 1932) and entropy regularized optimal transport (Léonard, 2013; Mikami & Thieullen, 2006; Mikami, 2004;

Léonard, 2010), a natural step to address this challenge in a simulation-free manner is to seek an extension to the OT-FM objective. Tong et al. (2023a) recently demonstrated the feasibility of just this approach through generalizing Eq. (2) and Eq. (3) to simultaneously regress on the conditional drift and score of an SDE. The proposed method replaces the flow ODE with the general Itô SDE:

$$dx = u_t(x)dt + g(t)dw_t, \quad x_0 \sim p_0(x_0) \tag{8}$$

where $u_t(x)$ is the SDE drift, $dw_t$ is standard Brownian motion[2], and $g(t)$ is a handcrafted function frequently taken to be constant (Tong et al., 2023a). The SDE drift and a corresponding ODE vector field $\hat{u}_t(x)$ are intimately related by the expression

$$\hat{u}_t(x) = u_t(x) - \frac{g(t)^2}{2} \nabla \log p_t(x) \tag{9}$$

denoted as the *probability flow* ODE of the process (Song et al., 2021), such that specification of the probability flow and *Stein score* function are sufficient to describe the SDE (Tong et al., 2023a).

Mirroring the previous flow-based models, the vector field, $u_t(x)$, and the score function, $s_t(x) = \nabla \log p_t(x)$, which define the SDE's drift are unknown. Tong et al. (2023a) propose a generalization of the FM objective, Eq. (3), for training approximators of both objects

$$\mathcal{L}_{[\text{SF}]^2\text{M}}(\theta) = \mathcal{L}_{\text{CFM}}(\theta) + \mathbb{E}_{t,q(z),p(x|z)} \lambda(t)^2 ||s_\theta(x,t) - \nabla \log p_t(x|z)||^2 \tag{10}$$

where $\lambda(t)^2$ is selected to standardize the loss, such that values of $\nabla \log p_t(x|z)$ near $t = 0$ or $t = 1$ do not dominate, effectively stabilizing training (Tong et al., 2023a; Song et al., 2021; Ho et al., 2020; Karras et al., 2022). Theorem 3.1 of their work demonstrates the feasibility of extending the marginalization construction in Eq. (2) to the marginal score through matching of the conditional score of Brownian bridge probability paths. Further background can be found in Appendix C.

In modeling the dynamics of real world systems, such score and flow matching models have displayed improved performance over OT-FM and FM (Tong et al., 2023a), in part due to the score function reducing movement to sparse regions of the data manifold. We emphasize that the conceptual similarity between the derivation of the flow matching objective and the generalized score and flow matching objective signifies that any improvements in the first can be readily transferred to the second without significant modification. As a result, in this work we theoretically restrict ourselves to purely flow-based models to simplify communication of novel developments. However, in later case studies we utilize score and flow-based models.

## 3 CONDITIONAL VARIABLE FLOW MATCHING

In the following section, we generalize the FM objective in Eq. (3) to matching the flow between arbitrary conditional distributions, $p_0(x|y)$ to $p_1(x|y)$, provided *unpaired* observations with distinct conditioning variables. We call our proposed framework *conditional variable flow matching* (CVFM).

### 3.1 CONSTRUCTING CONDITIONAL PROBABILITY PATHS AND VECTOR FIELDS

We begin by first noting that the original marginalization motivating the flow matching objective in Eq. (2) can be extended to construct a probability flow across both $x$ and a conditioning variable $y$ as

$$p_t(x|y) = \int p_t(x|y, z, w) q(z, w) dz dw \tag{11}$$

where $q(z, w)$ now denotes the empirical distribution over $z = (x_0, x_1)$ and $w = (y_0, y_1)$. We make the further assumption that the conditional joint probability path decomposes as $p_t(x, y|z, w) = p_t(x|z)p_t(y|w)$, resulting in two simultaneous conditional flows.

---

[2]The Brownian motion differential is defined to be standard Gaussian noise times a time differential.

We can also extend this line of thought towards defining a marginal conditional vector field, through marginalizing over vector fields conditioned on observations $z$ and $w$ as

$$u_t(x|y) = \mathbb{E}_{q(z,w)} \frac{u_t(x|z)p_t(x|z)p_t(y|w)}{p_t(x,y)} \tag{12}$$

where $u_t(x|z) : \mathbb{R}^N \to \mathbb{R}^N$ is a conditional vector field generating $p_t(x|z)$ from $p_0(x|z)$, without any explicit dependence upon the conditional distribution over our conditional variable, $y$. Following Theorem 3 (Lipman et al., 2023), Theorem 2.1, (Albergo & Vanden-Eijnden, 2023), and Theorem 3.1 (Tong et al., 2023a), we prescribe a form to both conditional vector fields such that they generate their respective conditional probability distributions[3]. Somewhat paradoxically, this way of combining conditional vector fields can be shown to generate the marginal conditional vector field, $u_t(x|y)$, which is formalized in the following theorem.

**Theorem 3.1** *The marginal conditional vector field Eq. (12) generates the marginal conditional probability path Eq. (11) from $p_0(x|y)$ given any samples of $q(z,w)$ independent of $x$, $y$, and $t$ if $q(z,w)$ follows the conditional optimal coupling $\pi(y_0, y_1)$ over $w$, and $q(y_0) = q(y_1)$.*

This result deviates from prior results by Lipman et al. (2023) (Theorem 1) and Albergo et al. (2023) (Theorem 2.6), showing that optimal transport over the conditioning variable is necessary for learning amortized conditional vector fields Chemseddine et al. (2024). The full proof of all theorems are provided in Appendix A.

### 3.2 FLOW MATCHING FOR CONDITIONAL DISTRIBUTIONS

Even provided Eq. (11) and Eq. (12), their incorporation in an overall objective for training a neural network approximator to $u_t(x|y)$ is still limited by several intractable integrals. Instead, we can obtain an unbiased estimator of the marginal conditional vector field and resulting probability path provided only samples from known distributions and the ability to compute $u_t(x|z)$ through the proposed conditional variable flow matching objective.

$$\mathcal{L}_{\text{CVFM}}(\theta) = \mathbb{E}_{t,q(z,w),p_t(x|z)p_t(y|w)} \left[ \alpha(w)\|v_\theta(x,y,t) - u_t(x|z)\|^2 \right] \tag{13}$$

This is formalized in the following recognizable theorem.

**Theorem 3.2** *If $p_t(x|y) > 0$ for all $x \in \mathbb{R}^N$ and for all $y \in \mathbb{R}^M$ and $t \in [0,1]$, then $\mathcal{L}_{MCFM}(\theta)$ (r.h.s. below) and $\mathcal{L}_{CVFM}(\theta)$ are equal up to a constant, and hence:*

$$\nabla_\theta \mathcal{L}_{\text{CVFM}}(\theta) = \nabla_\theta \mathbb{E}_{t,p_t(x,y)} \|v_\theta(x,y,t) - u_t(x|y)\|^2 \tag{14}$$

### 3.3 STABILIZING AND ACCELERATING TRAINING

Unlike previous flow matching frameworks, the empirical distribution, $q(z,w)$, cannot be arbitrary (Lipman et al., 2023; Tong et al., 2023a). Instead, the distribution must be selected so that movement from $q(y_0)$ to $q(y_1)$ follows the coupling $\pi(y_0, y_1)$. The theoretical reasons for this restriction are described in Appendix A. Although the construction of $q(z|w)$ is arbitrary, the conditional coupling $\pi(z|w)$ provides reduced objective variance and quicker convergence characteristics. Altogether, we recommend two changes to stabilize training.

**Conditional Optimal Transport**: We modify the empirical distribution, $q(z,w)$, through identifying a static conditional OT map: $q(z,w) = \pi(z,w)$ between the source and target distributions. Necessarily, the identification of this map requires the introduction of a ground cost with support over the space $\mathcal{X} \times \mathcal{Y} : \mathbb{R}^N \times \mathbb{R}^M$.

---

[3]For Gaussian probability paths with deterministic dynamics: $u_t(x|z) = x_1 - x_0$ (Pooladian et al., 2023; Tong et al., 2023b; Albergo & Vanden-Eijnden, 2023) and stochastic dynamic: $\hat{u}_t(x|z) = ((1-2t)/(2t(1-t)))(x - (tx_1 + (1-t)x_0)) + (x_1 - x_0)$, $\nabla_x \log p_t(x|z) = (tx_1 + (1-t)x_0 - x)/(\sigma^2 t(1-t))$ (Tong et al., 2023a).

We would like to search for an OT map predominantly permitting movement across $\mathcal{X}$ and not $\mathcal{Y}$. In a very practical sense, given the continuous support of $\mathcal{Y}$, such a constraint would not be feasible within a finite number of samples. Instead, we moderate this requirement in a manner similar to concurrent proposals by Kerrigan et al. (2024) and Chemseddine et al. (2024) in the form of the proposed continuous non-negative cost function

$$c((x_i, y_i), (x_j, y_j)) = \|x_i - x_j\|_p + \eta\|y_i - y_j\|_p \tag{15}$$

where $\eta > 0$ is a parameter governing the tolerance of transport permissible in $\mathcal{Y}$, and $\|x\|_p$ the $p$-norm .

**Proposition 3.1** *Let $\mu, \nu \in \mathcal{P}(\mathbb{R}^N \times \mathbb{R}^M)$ and let $\pi_\eta$ be an OT plan with associated value $\eta$ from the cost function Eq. (15). As $\eta \to \infty$, mass transport in $y$ is eliminated (Carlier et al., 2008):*

$$\lim_{\eta \to \infty} \int_{\mathbb{R}^M} \|y_i - y_j\|_p d\pi_\eta = 0 \tag{16}$$

**Conditioning Mismatch**: To further ensure that conditional OT is obtained within minibatches, we include a scaling term through a stationary symmetric kernel $\alpha(w)$ in Eq. (13). The kernel dictates the acceptance of conditional vector fields, dependent upon the degree of mismatch in the sampled conditioning variable $\{y_0, y_1\}$ – controlling the conditional probability flow permissible across $y$. In settings with discrete conditioning, we let it approach a delta function, prohibiting movement across classes, whereas with continuous conditioning and mild assumptions of continuity across $y$, providing some degree of relaxation is particularly advantageous. In this work, we restrict ourselves to the squared exponential kernel, $\alpha(w) = \exp(-(y_0 - y_1)/2\sigma_y^2)$ and the modulation of $\sigma_y$ (Wilson & Adams, 2013; Rasmussen & Williams, 2006). We will repeatedly see that this addition significant improves model performance and training stability.

## 4 EXPERIMENTS

We empirically evaluate the performance of the proposed CVFM framework on a suite of increasingly demanding problems, investigating its performance in accurately recovering target conditional densities. We first interrogate the performance of CVFM alongside conditional alternate formulations on 2D toy datasets with discrete and continuous conditioning. Subsequently, we turn towards domain transfer between MNIST and FashionMNIST, demonstrating high-dimensional conditional OT. Lastly, we model the dynamics of materials' microstructures undergoing spinodal decomposition across varied processing conditions.

**2D Experiments**: We first evaluate the capabilities of various methods in approximating dynamic conditional optimal transport and the Shrödinger bridge (SB) problem in a low-dimensional setting. We compare our complete method, CVFM, against Conditional Generative Flow Matching (CGFM) (Isobe et al., 2024; Zheng et al., 2023; Dao et al., 2023), Triangular Conditional Optimal Transport (T-COT-FM) (Kerrigan et al., 2024), as well as two ablated CVFM variants. CGFM in particular does not support *unpaired* conditioning, requiring the ability to sample from $z \sim \{\pi(x_0, x_1|y_i)\}_{i=0}^m$, precluding its ability to address scientific questions as $m \to \infty$. We simply present it as a useful benchmark, representing a lower-bound on the discrete conditioning problems presented. The first of the ablated variants considered removes both minibatch conditional optimal transport and $\alpha(w)$, but retains the interpolated flow in $y$ (see Appendix E). We refer to this variant as Conditional Flow Matching (CFM) as it reduces to the original proposed CFM algorithm given paired conditioning (Lipman et al., 2023; Albergo & Vanden-Eijnden, 2023). The second variant solely excludes $\alpha(w)$ – we refer to this as Conditional Optimal Transport Flow Matching (COT-FM). COT-FM is nearly equivalent to the concurrent work of Chemseddine et al. (2024); it differs by having varied noise schedules for the conditional probability paths in $x$ and $y$ (see Fig. 4). ODE and SDE-based variants for the SB problem are also evaluated for CVFM and COT-FM, utilizing the probability flow ODE with entropic regularized OT, alongside the SDE extension to our objective, conditional variable score and flow matching (CVSFM).

Three mappings are investigated and their associated results are displayed in Table 1. We report the Wasserstein-2 error in the predicted distribution at $t = 1$ to the target distribution, alongside

Table 1. Comparison of conditional neural optimal transport and Shrödinger bridge methods. Reported metrics consist of the Wasserstein-2 error between the target distribution and simulated distribution at $t = 1$, and the normalized path energy of the time-evolving distribution. Training was repeated over 5 seeds, with values reported as $\mu \pm \sigma$. Best observed values with *unpaired conditioning* are bolded with second best denoted by an asterisk within OT and SB methods. *Entropic* OT couplings, Eq. (5) identified via the Sinkhorn algorithm are differentiated from default *Exact* couplings, Eq. (4).

| | Wasserstein-2 Error ($\downarrow$) | | | Normalized Path Energy ($\downarrow$) | | |
| --- | --- | --- | --- | --- | --- | --- |
| | 8 Gaussian-8 Gaussian | 8 Gaussian-Moons | Moons-Moons | 8 Gaussian-8 Gaussian | 8 Gaussian-Moons | Moons-Moons |
| CVFM | **0.571±0.139** | **0.440±0.097** | **1.102±0.047** | 0.109±0.039* | **0.043±0.049** | **0.125±0.003** |
| COT-FM | 1.997±0.528* | 0.537±0.115* | 1.353±0.022 | 0.177±0.081 | 0.075±0.031* | 0.135±0.004* |
| CFM | 4.013±1.026 | 1.768±0.448 | 2.227±0.264 | 0.403±1.688 | 0.148±0.287 | 0.410±0.061 |
| T-COT-FM | 2.296±0.060 | 0.564±0.031 | 1.318±0.041* | **0.063±0.024** | 0.081±0.006 | 0.209±0.023 |
| CVFM-Entropic | **0.409±0.080** | **0.414±0.109** | **1.018±0.031** | 0.064±0.040 | 0.022±0.037 | 0.088±0.013 |
| COT-FM-Entropic | 2.566±0.624 | 0.720±0.176 | 1.080±0.016* | 0.179±0.073 | 0.073±0.064 | 0.125±0.004 |
| CVSFM | 0.518±0.105* | 0.428±0.114* | 1.081±0.028 | 0.089±0.033* | 0.041±0.056* | 0.115±0.013 |
| COT-SFM | 1.936±0.423 | 0.539±0.133 | 1.267±0.043 | 0.159±0.085 | 0.064±0.049 | 0.110±0.013* |
| CGFM | 0.491±0.095 | 0.352±0.085 | – | 0.024±0.038 | 0.019±0.054 | – |
| CGFM-Entropic | 0.483±0.075 | 0.376±0.083 | – | 0.037±0.066 | 0.019±0.065 | – |

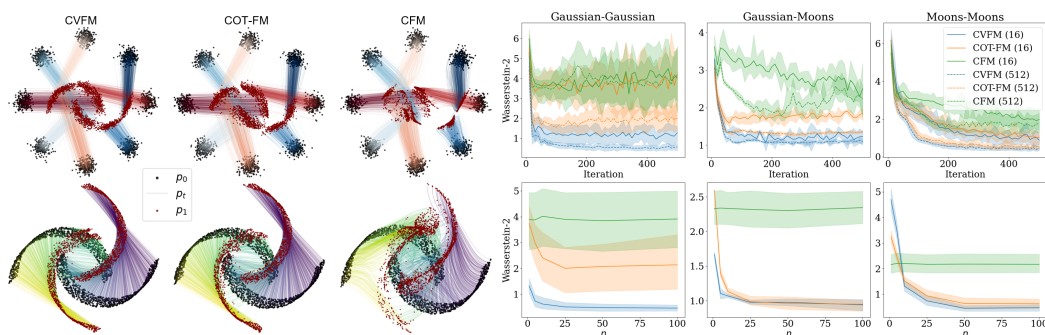

Figure 2. CVFM results in lower error in Wasserstein-2 distance to target distribution across batch sizes and $y$-direction cost weighting $\eta$, compared to COT-FM (Eq. (15)) or the naïve conditional implementation CFM. Trajectories are colored by conditioning variable.

the normalized path energy, defined as $\text{NPE}(\mu, \nu) = |\int \|v_\theta(x, y, t)\|^2 dt - W_2^2(\mu, \nu)|/W_2^2(\mu, \nu)$, computed through 10,000 samples. The CVFM method notably results in equivalent or more optimal values of Wasserstein-2 error and normalized path energy across all mappings with unpaired conditioning. CVFM even nearly matches CGFM's performance on discrete problems, where CGFM has knowledge of the correct conditional couplings *a priori*. Figure 2 contrasts samples and pathways from a subset of these methods. Only CVFM is able to clearly recreate the final target distributions. Notably, the CFM formulation, produces an appreciable nonzero vector field in the conditioning space, precluding the validity of Eq. (12) (Appendix A), resulting in poor performance. Noteably, the kernel appreciably improves performance; CVFM outperforms both T-COT-FM and COT-FM.

**Improved convergence**: Beyond improved metrics, the CVFM implementation also displays improved convergence characteristics, as evidenced by Figure 2. In all cases, the CVFM objective converges faster and to a lower metric value than other methods and variants. Further interrogation regarding the impact of $\eta$ in the conditional ground cost and $\alpha(w)$ was also performed. In the proposed methods incorporating the ground cost (CVFM, COT-FM, and T-COT-FM), increasing $\eta$ leads to improvement. However, it is often insufficient. For example, in the *8 Gaussian - 8 Gaussian* example the addition of $\alpha(w)$ in CVFM facilitates a marked improvement. Additional ablations are also presented in Appendix D.1 highlighting the important role of $\alpha(w)$ in reducing the COT approximation inherent in Eq. (15) in greater depth.

**MNIST-FashionMNIST Domain Transfer**: We further evaluate our method in domain transfer between MNIST digits and FashionMNIST clothing articles with conditioning by class. We directly compare our two variants, CVSFM against COT-SFM, Figure 3. The FID and LPIPS scores reported

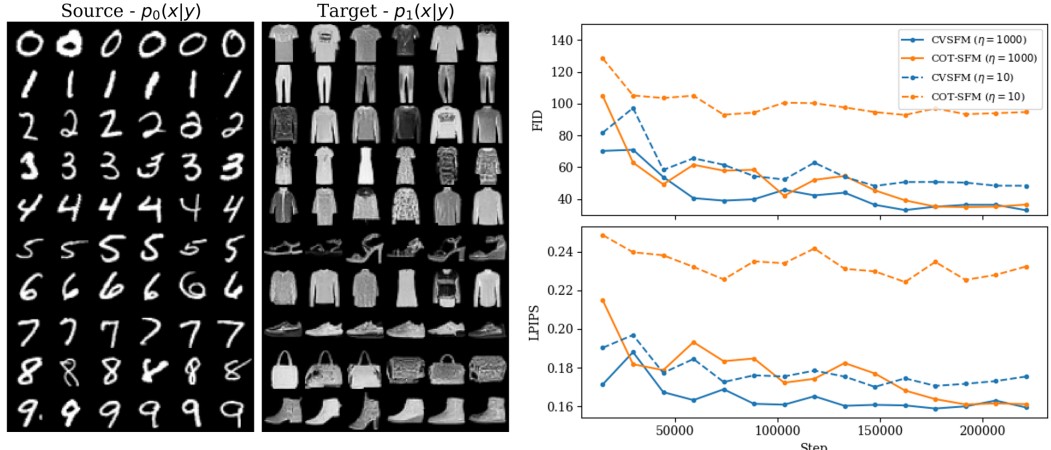

Figure 3. CVSFM conditionally generated images from the FashionMNIST dataset. (Left) pane displays samples from the initial distributions $p_0(x|y)$, while the middle displays generated samples from $p_1(x|y)$. Relative positioning indicates paired samples. (Right) pane illustrates the improved convergence of CVSFM over COT-SFM in high-dimensional domain transfer for $\eta = 10$ and $\eta = 1000$. Displayed FID/LPIPS scores are computed per class and averaged.

are averages across the class conditioned scores (i.e., FID and LPIPS are computed on a per-class basis across 1,000 samples in each class). Select images from the source and corresponding generated images from the target conditional distributions are also shown. As seen in the repeated samples, the model is able to consistently map to the correct paired conditional distribution while displaying appreciable diversity within each class. Similar to the 2D experiments, we observe improved convergence and mode coverage with CVSFM across $\eta$ values due to $\alpha(w)$ restricting flow across $y$. For further analysis, see Appendix D.2.

**Material Dynamics**: We next investigate the performance of our proposed method in a scientific application: the dynamics of time-evolving microstructures subject to various processing conditions. Low-dimensional representations of the 2-point statistics[4] of phase-field simulations are taken as descriptors of the internal state of the material's microstructure, discussed in greater detail in Appendix B. Importantly, the time-dependent observations modeled are the result of computational simulations of spinodal decomposition, granting us complete access to *paired* conditioning-trajectory information – information frequently unavailable in manufacturing settings. This enables us to validate and quantify the model's performance by computing error metrics on a per trajectory basis. Table 2 reports the results for our proposed methodology, accepting *unpaired* samples and conditioning variables in comparison with traditional approaches requiring complete trajectory information. Despite having access to corrupted and *misaligned* versions of the available observations, CVSFM-Exact results in comparable or better performance when compared to the Neural ODE or LSTM models which require complete trajectories. We also compare our method against T-COT-FM (Kerrigan et al., 2024), similarly capable of accepting corrupted conditional measurements, alongside two SDE-based extensions, denoted T-COT-SFM. The first utilizes a fixed value of $\sigma$, while in the second, we extend the work of Kerrigan et al. (2024) to set a distinct conditional probability path across $y$, with its own $\sigma_y$ as implemented in this work. Figure 4 displays exemplar trajectories alongside distributions of expected error across time, further illustrating the ability of the proposed method to learn complex conditional stochastic nonlinear dynamics.

## 5 CONCLUSIONS

We have proposed a novel framework, capable of learning to transform conditional distributions between general source and target distributions given *unpaired* samples. With the same underlying approach, we have also presented extensions capable of approximating the conditional Schrödinger

---

[4]Compression is performed through Principal Component Analysis (PCA).

Table 2. Comparison of conditional neural optimal transport methods alongside conventional approaches on spinodal decomposition PC trajectories. Expected values of $\mathbb{E}_t[p_t(x|y)]$ are compared against complete trajectory information not observed during training. Reported absolute error metrics are reported as $\mu \pm \sigma$ alongside maximum and minimum absolute error.

| | Train | | | Test | | |
|---|---|---|---|---|---|---|
| | $\mu \pm \sigma$ | Min. | Max. | $\mu \pm \sigma$ | Min. | Max. |
| Neural ODE (Chen et al., 2018) | 0.261±0.119 | 0.105 | **1.499** | 0.264±0.129 | 0.116 | 1.400 |
| LSTM (Hochreiter & Schmidhuber, 1997) | 0.355±0.212 | 0.038 | 2.071 | 0.358±0.331 | 0.045 | 1.648 |
| CVSFM | **0.166±0.140** | **0.023** | 1.654 | **0.188±0.147** | 0.039 | **1.284** |
| CVSFM-Entropic | 0.378±0.322 | 0.045 | 2.112 | 0.388±0.320 | 0.044 | 2.032 |
| COT-SFM | 0.202±0.348 | 0.034 | 11.422 | 0.218±0.489 | **0.033** | 11.649 |
| COT-SFM-Entropic | 0.524±0.306 | 0.108 | 2.568 | 0.531±0.307 | 0.110 | 2.012 |
| T-COT-FM (Kerrigan et al., 2024) | 0.899±0.370 | 0.163 | 3.47 | 0.901±0.365 | 0.161 | 2.639 |
| T-COT-SFM | 0.636±0.279 | 0.235 | 2.331 | 0.637±0.282 | 0.194 | 2.335 |
| T-COT-SFM ($\sigma_y$) | 0.332±0.223 | 0.073 | 2.525 | 0.329±0.222 | 0.075 | 1.786 |

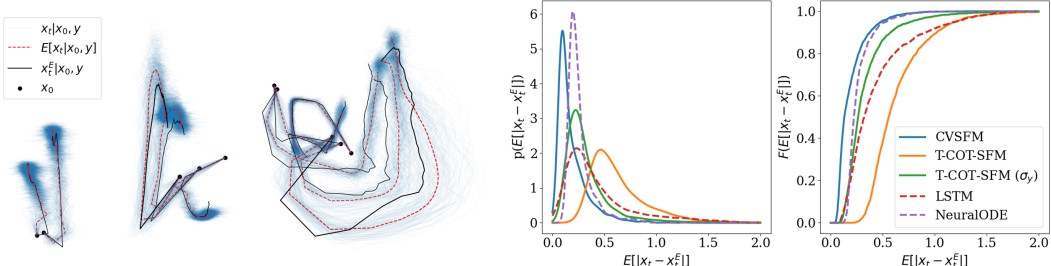

Figure 4. Collection of 5 randomly sampled trajectories from the test set in projections of PC1-PC2 and PC2-PC3 displaying (left) 128 samples in blue from CVSFM-Exact, with the expected value in red, and (right) probability density and cumulative distribution function of absolute error compared against alternate approaches.

bridge problem. Our central contribution is a novel algorithmic framework for measures in a conditional Wasserstein space, equipped with a regularized conditional distance metric, an independent conditioning variable flow, and a kernel enforcing structure across the learned vector fields in the conditioning variable. We verify our proposed approach through synthetic and real-world tasks, demonstrating notable improvements over prior methods, and validating the feasibility of learning conditional dynamical processes from unaligned measurements.

REPRODUCIBILITY

We have included several appendices describing in detail the theoretical derivations referenced in the paper (Appendix A) as well as describing the proposed algorithm (Appendix E) and the experiments (Appendix B, C, D, and E).

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

# A  APPENDIX: PROOFS OF THEOREMS

## A.1  PROOF $p_t(x|y)$ AND $u_t(x|y)$ SATISFY THE CONTINUITY EQUATION

**Theorem 3.1** *The marginal conditional vector field Eq. (12) generates the marginal conditional probability path Eq. (11) from $p_0(x|y)$ given any samples of $q(z, w)$ independent of $x$, $y$, and $t$ if $q(z, w)$ follows the conditional optimal coupling $\pi(y_0, y_1)$ over $w$, and $q(y_0) = q(y_1)$.*

*Proof.* The continuity equation provides a necessary and sufficient condition for a vector field to generate a probability distribution (Villani, 2009). Therefore, the proof is completed by demonstrating that $u_t(x|y)$, defined by Eq. (12), meets the continuity equation for the conditional distribution $p_t(x|y)$. We utilize the assumed decomposition introduced in the main body of the paper, $p(x, y|z, w) = p(x|z)p(y|w)$. Additionally, we assume that $x$ is independent of $w$ given $y$, $p_t(x|y, w) = p_t(x|y)$. Finally, we will require that $p(y_0) = p(y_1)$. In other words, that the distribution on the conditioning variable is the same at $t_0$ and $t_1$. We argue that this is not a very restrictive requirement in practice. It is the situation that occurs in many applications, in particular scientific dynamics problems where the distribution on conditions is constant over all time.

$$\frac{d}{dt}\left(p_t(x|y)\right) = -\text{div}_x(p_t(x|y)u_t(x|y)) \tag{A.1}$$

We begin with expanding the left hand side of the continuity equation in Eq. (A.1). To clarify notation, we utilize $p(\cdot)$ to denote probability density functions over the primary variables, $x$ and $y$. We use $q(\cdot)$ to denote distributions over the pair variables $z$ and $w$. The subscript $t$ denotes time dependence. Their arguments differentiate the multiple distributions of each type.

$$\frac{d}{dt}\left(p_t(x|y)\right) = \frac{d}{dt}\left(\frac{p_t(x, y)}{p_t(y)}\right)$$
$$= \frac{d}{dt}\left(\int \frac{p_t(x, y|w, z)}{p_t(y)}q(w, z)dzdw\right)$$

We utilize the following assumed decomposition, $p_t(x, y|w, z) = p_t(x|z)p_t(y|w)$,

$$= \frac{d}{dt} \left( \int \frac{p_t(x|z)p_t(y|w)}{p_t(y)} q(w,z) dz dw \right)$$

$$= \int \left[ \underbrace{\frac{p_t(y|w)}{p_t(y)} \frac{d}{dt}(p_t(x|z))}_{T_1} - \underbrace{\frac{p_t(x|z)p_t(y|w)}{p_t(y)^2} \frac{d}{dt}(p_t(y))}_{T_2} + \underbrace{\frac{p_t(x|z)}{p_t(y)} \frac{d}{dt}(p_t(y|w))}_{T_3} \right] q(w,z) dz dw$$

We next consider each of the individual terms above. We will extensively rely upon the fact that we have prescribed forms to $u_t(x|z)$ and $u_t(y|w)$ such that they generate $p_t(x|z)$ and $p_t(y|w)$, respectively (see Theorem 3 of Lipman et al. (2023) or Theorem 2.1 in Tong et al. (2023b)). The integrands are further assumed to satisfy the regularity conditions of the Leibniz Rule for changing the order of integration and differentiation. Beginning with the first term, $T_1$,

$$T_1 = \int \frac{p_t(y|w)}{p_t(y)} \frac{d}{dt}(p_t(x|z)) q(w,z) dz dw$$

as $u(x|z)$ generates $p(x|z)$,

$$= -\int \frac{p_t(y|w)}{p_t(y)} \mathrm{div}_x(p_t(x|z)u_t(x|z)) q(w,z) dz dw$$

Leibniz Rule,

$$= -\mathrm{div}_x \left( \int \frac{p_t(y|w)p_t(x|z)q(w,z)}{p_t(y)} u_t(x|z) dz dw \right)$$

$$= -\mathrm{div}_x \left( p_t(x|y) \int \frac{p_t(y|w)p_t(x|z)q(w,z)}{p_t(y)p_t(x|y)} u_t(x|z) dz dw \right)$$

using the definition of $u_t(x|y)$, Eq. (12),

$$= -\mathrm{div}_x \left( p_t(x|y)u_t(x|y) \right)$$

Here, we arrive at the right hand side of the continuity equation. Next, we turn to demonstrating that terms $T_2$ and $T_3$ equate to zero. We begin inspecting the second term, $T_2$,

$$T_2 = \int -\frac{p_t(x|z)p_t(y|w)}{p_t(y)^2} \frac{d}{dt}(p_t(y)) q(w,z) dz dw$$

$$= -\frac{1}{p_t(y)^2} \frac{d}{dt}(p_t(y)) \int p_t(x|z)p_t(y|w) q(w,z) dz dw$$

$$= -\frac{p_t(x|y)}{p_t(y)} \frac{d}{dt} p_t(y)$$

Next, we consider the last term, $T_3$, before returning to $T_2$.

$$T_3 = \int \frac{p_t(x|z)}{p_t(y)} \frac{d}{dt}(p_t(y|w)) q(w,z) dz dw$$

as $u_t(y|w)$ generates $p_t(y|w)$,

$$= -\int \frac{p_t(x|z)}{p_t(y)} \text{div}_y (p_t(y|w)u_t(y|w))q(w,z)dzdw$$

Leibniz Rule,

$$= -\frac{1}{p_t(y)}\text{div}_y \left( \int u_t(y|w)p_t(x|z)p_t(y|w)q(w,z)dzdw \right)$$

marginalization on $z$, the chain rule of probability, and the second assumption outlined previously,

$$= -\frac{1}{p_t(y)}\text{div}_y \left( \int u_t(y|w)p_t(x|y)p_t(y|w)q(w)dw \right)$$

$$= -\frac{1}{p_t(y)}\text{div}_y \left( p_t(x|y)p_t(y) \int u_t(y|w)\frac{p_t(y|w)q(w)}{p_t(y)}dw \right)$$

definition of the marginal vector field, Eq. (8) in Lipman et al. (2023) or Eq. (9) in Tong et al. (2023b)

$$= -\frac{1}{p_t(y)}\text{div}_y (p_t(x|y)p_t(y)u_t(y))$$

product rule,

$$= -\frac{1}{p_t(y)}p_t(y)u_t(y)^T\nabla_y p_t(x|y) - \frac{p_t(x|y)}{p_t(y)}\text{div}_y (p_t(y)u_t(y))$$

$u_t(y)$ generates $p_t(y)$,

$$= -\frac{1}{p_t(y)}p_t(y)u_t(y)^T\nabla_y p_t(x|y) + \frac{p_t(x|y)}{p_t(y)}\frac{d}{dt}p_t(y)$$

$$= -u_t(y)^T\nabla_y p_t(x|y) + \frac{p_t(x|y)}{p_t(y)}\frac{d}{dt}p_t(y)$$

Combining $T_1$, $T_2$, and $T_3$ together, we obtain,

$$\frac{d}{dt}(p_t(x|y)) = T_1 + T_2 + T_3$$

$$= -\text{div}_x (p_t(x|y)u_t(x|y)) - \frac{p_t(x|y)}{p_t(y)}\frac{d}{dt}p_t(y) - u_t(y)^T\nabla_y p_t(x|y) + \frac{p_t(x|y)}{p_t(y)}\frac{d}{dt}p_t(y)$$

$$= -\text{div}_x (p_t(x|y)u_t(x|y)) - u_t(y)^T\nabla_y p_t(x|y)$$

**Optimal Transport between** $p(y_0) = p(y_1)$: The continuity equation is left with $-u_t(y)^T\nabla_y p_t(x|y)$. In general, this remainder is nonzero. For arbitrary $p(x_0, y_0)$ and $p(x_1, y_1)$, the $u_t(x|y)$ defined by Eq. (12) generates $p_t(x|y)$ only if the flow over $y$ is carefully designed, such that it is orthogonal to $\nabla_y p_t(x|y)$ for all $t$.

In fact, in the problem settings discussed in this paper and under the proposed CVFM training scheme, we are in one of these rare situations. Consider our final assumption: $p(y_0) = p(y_1)$. When $p(y_0) = p(y_1)$, the optimal transport flow does nothing, resulting in $dp_t(y)/dt = 0$ and $u_t(y) = 0$. Importantly, in CVFM, we only utilize a conditional flow, $u_t(y|w)$, which approximates the flow over $y$, $u_t(y)$, in expectation. In general, this type of conditional approximation is not guaranteed to produce the marginal optimal transport map. However, when static optimal transport is utilized in

training, Tong et al. (2023a); Pooladian et al. (2023) show that this results in the expectation of the conditional flow sufficiently approximating the marginal OT.

We directly observed the impact of minibatch OT and this remainder in early experiments. Without minibatch OT, the learned $u_t(x|y)$ cannot generate the required marginal conditional vector field generating $p_t(x|y)$. Only once conditional optimal transport was incorporated could the appropriate push-forward operation be obtained. Importantly, this behavior differs from previous flow matching efforts where learning the appropriate vector field is possible with or without optimal transport (Albergo et al., 2023; Pooladian et al., 2023; Liu et al., 2022b; Tong et al., 2023a).

Therefore, because $p(y_0) = p(y_1)$ and static optimal transport is used in training, Section 3.3, the remainder disappears and $u_t(x|y)$ defined by Eq. (12) and $p_t(x|y)$ satisfy the continuity equation.

$$\frac{d}{dt}\left(p_t(x|y)\right) = -\text{div}_x\left(p_t(x|y)u_t(x|y)\right) - \underbrace{u_t(y)^{T}}_{}\overbrace{\nabla_y p_t(x|y)}^{0}$$

$$= -\text{div}_x\left(p_t(x|y)u_t(x|y)\right)$$

## A.2 EQUIVALENCE OF THE FLOW MATCHING OBJECTIVE

**Theorem 3.2** *If $p_t(x|y) > 0$ for all $x \in \mathbb{R}^N$ and for all $y \in \mathbb{R}^M$ and $t \in [0,1]$, then, $\mathcal{L}_{MCFM}(\theta)$ (r.h.s. below) and $\mathcal{L}_{CVFM}(\theta)$ are equal up to a constant, and hence:*

$$\nabla_\theta \mathcal{L}_{\text{CVFM}}(\theta) = \nabla_\theta \mathbb{E}_{t,p_t(x,y)}||v_\theta(x,y,t) - u_t(x|y)||^2 \tag{A.2}$$

*Proof.* As in Lipman et al. (2023), several assumptions are necessary to guarantee the existence of various intergrals and to allow exchanging of their order. Specifically, we assume $q(w,z)$, $p_t(x|z)$, $p_t(y|w)$ and $p(w)$ decrease to zero as $||x||, ||y|| \to \infty$ and that $v_t$, $v_t$, and $\nabla_\theta v_t$ are bounded. Expectations taken relative to $\pi(z,w)$ as $\eta \to \infty$ results in $\alpha(w) \to 1$.

We begin by stating the intractable marginal conditional flow matching (MCFM) objective

$$\mathcal{L}_{MCFM}(\theta) = \mathbb{E}_{t,p_t(x,y)}||v_\theta(x,y,t) - u_t(x|y)||^2 \tag{A.3}$$

Expanding the $L_2$-norm

$$\mathcal{L}_{MCFM}(\theta) = \mathbb{E}_{t,p_t(x,y)}||v_\theta(x,y,t) - u_t(x|y)||^2$$

$$= \mathbb{E}_{t,p_t(x,y)}\left[\underbrace{||v_\theta(x,y,t)||^2}_{T_1} - \underbrace{2\langle v_\theta(x,y,t), u_t(x|y)\rangle}_{T_1} + \underbrace{||u_t(x|y)||^2}_{T_3}\right]$$

Consider each term independently. Note, the third term can be ignored because it is independent of the trainable parameters.

$$T_1 = \mathbb{E}_{t,p_t(x,y)}\left[||v_\theta(x,y,t)||^2\right]$$

$$= \mathbb{E}_t\left[\int p_t(x,y,w,z)||v_\theta(x,y,t)||^2 dzdwdxdy\right]$$

$$= \mathbb{E}_t\left[\int p_t(x|z)p_t(y|w)q(z,w)||v_\theta(x,y,t)||^2 dzdwdxdy\right]$$

$$= \mathbb{E}_{t,p_t(x|z),p_t(y|w),q(z,w)}\left[||v_\theta(x,y,t)||^2\right]$$

Consider the second term.

$$T_2 = -2\mathbb{E}_{t,p_t(x,y)}\left[\langle v_\theta(x,y,t), u_t(x|y)\rangle\right]$$

$$= -2\mathbb{E}_t\left[\int p_t(x,y)\left\langle v_\theta(x,y,t), \int u_t(x|z)\frac{p_t(x,y|z,w)q(z,w)}{p_t(x,y)}dzdw\right\rangle dxdy\right]$$

$$= -2\mathbb{E}_t\left[\int p_t(x,y)\frac{p_t(x,y|z,w)q(z,w)}{p_t(x,y)}\langle v_\theta(x,y,t), u_t(x|z)\rangle dxdydzdw\right]$$

$$= -2\mathbb{E}_t\left[\int p_t(x|z)p_t(y|z)q(z,w)\langle v_\theta(x,y,t), u_t(x|z)\rangle dxdydzdw\right]$$

$$= -2\mathbb{E}_{t,p_t(x|z),p_t(y|z),q(z,w)}\left[\langle v_\theta(x,y,t), u_t(x|z)\rangle\right]$$

Combining these together and comparing them against the expanded form for $\mathcal{L}_{CVFM}(\theta)$, we clearly see that the $\mathcal{L}_{MCFM}(\theta)$ and $\mathcal{L}_{CVFM}(\theta)$ objectives are equivalent up until a constant independent of the training parameters, $\theta$.

### A.3 CONTINUITY ACROSS CONDITIONAL VECTOR FIELDS

**Theorem 3.3** *If $v_\theta(x,y,t)$ is locally Lipschitz continuous, $p_t(y|w), p_t(x|z) \in \mathcal{C}^\infty$, and $\mathcal{L}_{CVFM}(\theta)$ is continuous and differentiable with respect to $y$, then the learned network $v_\theta(x,y,t)$ generates conditional probability paths across the conditional Wasserstein density manifold.*

*Proof.* By the composition theorem, the composition of continuous functions results in a function which is similarly continuous.

We begin with the empirical distribution $q(z,w)$, which as a finite sum of Dirac delta functions is itself not inherently continuous, although expectations taken with respect to $q(z,w)$ are. Also by construction $p_t(y|w), p_t(x|z)$ are smooth and continuous distributions. $\mathcal{L}_{CVFM}(\theta)$ (Eq. (3)) is a continuous function of $v_\theta(x,y,t)$, $\alpha(w)$, and $u_t(x|z)$ by which expectations over $q(z,w), p_t(y|w), p_t(x|z)$ maintain this property.

Empirical samples are drawn according to the OT coupling $\pi(z,w)$, minimizing the ground cost defined in Eq. (15), and inducing a gradient flow in the Wasserstein sense. Section A.1 previously established adherence to the continuity equation.

Therefore, the learned network $v_\theta(x,y,t)$ results in a continuous vector field across $y$, generating $p_t(x|y)$ which respects the underlying geometry of the Wasserstein density space.

## B APPENDIX: CASE STUDY BACKGROUND

### B.1 MICROSTRUCTURE EVOLUTION IN MATERIALS INFORMATICS

**Phase-Field modeling**: Phase-field simulations are commonly applied to model a number of manufacturing processes involving evolving interfaces (such as solidification/melting, spinodal decomposition, grain growth, recrystallization, and crack propagation (Steinbach, 2009; Miehe et al., 2010)). In particular, the Cahn-Hilliard equation is frequently used to describe spinodal decomposition, a spontaneous thermodynamic-instability-induced phase separation (Cahn & Hilliard, 1958); this partial differential equation models a diffusion-driven process with a diffusivity constant $\mathcal{D}$, driving the evolution of composition variations, $c$, over characteristic length scales dictated by a gradient energy coefficient, $\gamma$. In our problem the spatially-dependent composition will take the role of material microstructure.

$$\frac{\delta c}{\delta t} = \mathcal{D}\nabla^2\left(c^3 - c - \gamma\nabla^2 c\right) \tag{B.1}$$

The dataset utilized in this paper's third presented case study was simulated using the MEMPHIS code base from Sandia National Labs (Dingreville et al., 2020). Details on the dataset are included in Appendix E.6.

**2-Point spatial correlations**: This work uses 2-point spatial correlations (Torquato, 2002; Kalidindi, 2015; Adams et al., 2013) as descriptive microstructure features. This representation builds on the idea that the microstructure itself is a stochastic function, where individual observed microstructure instances are simply samples from the governing stochastic microstructure function (Kröner, 1971; Torquato, 2002). Within this conceptualization, features – such as the 2-point statistics – are designed to quantify the governing stochastic microstructure function since it is specifically the governing stochastic microstructure function that dictates the properties and behaviors of a sampled microstructure. This theoretical treatment accounts for the inherent stochasticity displayed by material microstructures, and allows for the underlying stochastic function to be linked to homogenized properties. It also provides a convenient mechanism to account for underlying symmetries (such as translation-equivariance and periodicity).

In practice, 2-point spatial correlations can be computed as a convolution of the sampled discrete microstructure function $m_s^\alpha$, where $\alpha$ indexes the material local state and $s$ indexes the spatial voxel. The resulting 2-point spatial correlations between two arbitrary material states, $\alpha$ and $\beta$, are then defined by the operation

$$f_r^{\alpha\beta} = \frac{1}{S}\sum_{s=1}^{S} m_s^\alpha m_{s+r}^\beta \qquad \text{(B.2)}$$

where $S$ is the number of voxels in the microstructural domain. These represent lower-order terms in a moment expansion of the true microstructure random process; for a number of materials systems, this term is dominant and captures most of the variation in bulk material properties (Kalidindi, 2015). Dimensionality reduction techniques can then be effectively applied to a dataset of 2-point spatial correlations to provide robust, information-dense features for the construction of linkages between process parameters and internal material structure (Gupta et al., 2015; Latypov et al., 2019; Yabansu et al., 2020; Marshall & Kalidindi, 2021; Paulson et al., 2017; Generale & Kalidindi, 2021; Kalidindi, 2020; Harrington et al., 2022).

## C   APPENDIX: ADDITIONAL SIMULATION-FREE SCORE AND FLOW MATCHING BACKGROUND

Tong et al. (2023a) proposed a simulation-free training objective for approximating continuous time Schrodinger bridges (Bunne et al., 2023a; De Bortoli et al., 2023), generalizing Flow Matching (FM) (Lipman et al., 2023; Albergo & Vanden-Eijnden, 2023) to the case of stochastic dynamics with arbitrary source distributions. Let $p : [0,1] \times \mathbb{R}^N \to \mathbb{R}^+$ define a time-dependent probability path, $u : [0,1] \times \mathbb{R}^N \to \mathbb{R}^N$ a time-dependent vector field, and $g : [0,1] \to \mathbb{R}_{>0}$ a continuous positive diffusion function. An associated Itô stochastic differential equation (SDE) can be defined

$$dx = u_t(x)dt + g(t)dw_t \qquad \text{(C.1)}$$

where $u_t(x)$ is equivalent to $u(t,x)$, and $dw_t$ is standard Brownian motion. Utilizing the *Fokker-Planck equation* and *continuity equation*, it is possible to derive the *probability flow* ordinary differential equation (ODE) of the process (Tong et al., 2023a; Song et al., 2021) and comprise a relation between the probability flow ODE $\hat{u}_t(x)$ and the SDE drift as

$$u_t(x) = \hat{u}_t(x) + \frac{g^2(t)}{2}\nabla \log p_t(x). \qquad \text{(C.2)}$$

As long as the probability flow ODE and score function can be specified, the SDE can be adequately described. Tong et al. (2023a) demonstrated that the intuition underpinning Eq. (11) can also be extended to the marginalization over conditional scores, resulting in the expressions

$$\hat{u}_t(x) = \int \hat{u}_t(x|z)\frac{p_t(x|z)q(z)}{p_t(x)}dz, \quad \nabla \log p_t(x) = \int \nabla \log p_t(x|z)\frac{p_t(x|z)q(z)}{p_t(x)}dz. \qquad \text{(C.3)}$$

**Gaussian marginal conditional flows**: Prior works (Theorem 3 (Lipman et al., 2023), Theorem 2.1 (Tong et al., 2023b), Theorem 2.6, (Albergo & Vanden-Eijnden, 2023)) have demonstrated a method for tractably evaluating Eq. (C.3) in the case where the ODE/SDE conditional flows are Gaussian (i.e., $p_t(x|z) = \mathcal{N}(x; \mu_t(z), \sigma_t^2(z))$). The unique vector field $\hat{u}_t(x|z)$ generating this flow has the form

$$\hat{u}_t(x|z) = \frac{\sigma_t'(z)}{\sigma_t(z)}(x - \mu_t(z)) + \mu_t'(z) \tag{C.4}$$

where $\sigma_t'(z)$ and $\mu_t'(z)$ denote the time derivatives of $\sigma_t(z)$ and $\mu_t(z)$, respectively. This can seamlessly be extended to define the conditional score $\nabla \log p_t(x|z) = -(x - \mu_t(z))/\sigma_t^2(z)$. In the particular case of a *Brownian bridge* from $x_0$ to $x_1$, sampled from $q(z)$, with constant diffusion rate $g(t) = \sigma$, the conditional flow is defined as $p_t(x|z) = \mathcal{N}(x; tx_1 + (1-t)x_0, \sigma^2 t(1-t))$, resulting in

$$\hat{u}_t(x|z) = \frac{1 - 2t}{t(1-t)}(x - (tx_1 + (1-t)x_0)) + (x_1 - x0)$$

$$\nabla \log p_t(x|z) = \frac{tx_1 + (1-t)x_0 - x}{\sigma^2 t(1-t)}.$$

In a similar manner to the derivation shown in Appendix A, a density over initial conditions $p(x_0)$, induces marginal distributions $p_t(x)$ satisfying the *Fokker-Planck equation* where $\Delta p_t = \nabla \cdot (\nabla p_t)$ (Tong et al., 2023a):

$$\frac{\partial p}{\partial t} = -\nabla \cdot (p_t u_t) + \frac{g^2(t)}{2}\Delta p_t \tag{C.5}$$

In total, Tong et al. (2023a) demonstrate that the concept of regressing upon the conditional vector field from FM can be extended to regressing upon conditional drift and score, providing improved performance in practice.

**Weighting schedule** $\lambda(t)$: In the case with conditional Gaussian $p_t(x|z)$ probability paths, as in Eq. (C.4), Tong et al. (2023a) advocate a particular weighting schedule $\lambda(t)$:

$$\lambda(t) = \frac{2\sigma_t}{\sigma^2} = \frac{2\sqrt{t(1-t)}}{\sigma}. \tag{C.6}$$

This weighting schedule provides simplification to the objective alongside numerical stability, converting the score matching objective to

$$\lambda(t)^2 \|s_\theta(x,t) - \nabla_x \log p_t(x|z)\|^2 = \|\lambda(t)s_\theta(x,t) + \varepsilon\|^2 \tag{C.7}$$

where $\varepsilon \sim \mathcal{N}(0,1)$.

# D    APPENDIX: ADDITIONAL CASE STUDY SPECIFIC RESULTS

We move on towards presenting additional results obtained throughout this work. The results expand upon the discussion presented in Section 4 towards interrogating the empirical performance of CVFM, illuminating the critical advances necessary for learning conditional vector fields.

## D.1    2D EXPERIMENTS

In Figure 2 trajectories of the learned vector fields were presented for the *8 Gaussian - Moons* mapping with discrete conditioning and the *Moons - Moons* with continuous conditioning, although this remains a fraction of the cases run. For completeness, we present the trajectories of all methodologies evaluated in Table 1 in Figure D.1.

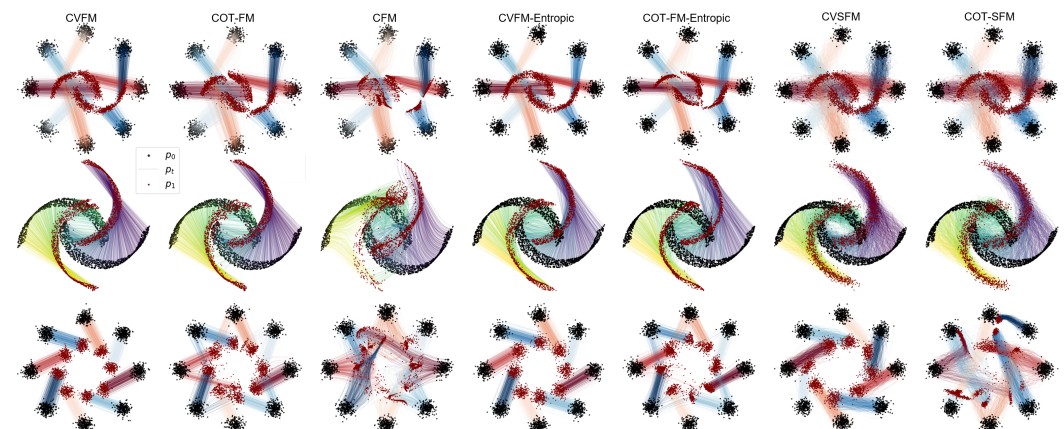

Figure D.1. Comparison of obtained trajectories for various OT and SB modeling approaches in the synthetic datasets considered associated with Wasserstein-2 error and normalized path energy reported in Table 1. Trajectories are colored by the conditioning variable.

While, perhaps visually the simplest, the *8 Gaussian - 8 Gaussian* mapping, consisting of a 45 degree rotation about the origin, is demonstrably the most complex family of conditional vector fields to learn given samples of the empirical distribution $q(w, z)$. This is the only case in which only CVFM variants reliably learn to disentangle the conditional dynamics (Figure D.1), while COT-FM attempts to split mass to minimize transport costs across the joint $\mathcal{X} \times \mathcal{Y} : \mathbb{R}^N \times \mathbb{R}^M$, visualized by splits mapping to target densities with similar conditioning values.

**Objective Target Variance**: One reason for the improved covergence of CVFM is the significantly lower variance of the training objective target in CVFM in comparison to COT-FM. We define the objective target variance (OTV) as

$$\text{OTV}_{\text{CVFM}} = \mathbb{E}_{t,q(z,w),p(w),p_t(x,y|z,w)} ||u_t(x|z)||^2$$
$$\text{OTV}_{\text{COT-FM}} = \mathbb{E}_{t,q(z,w),p_t(x,y|z,w)} ||u_t(x|z)||^2 \tag{D.1}$$

Notably, these equations do not include the true underlying conditional vector field and, as such, the computed values should not be compared across test cases. Figure D.2 demonstrates the stark contrast between COT-FM and CVFM on the *8 Gaussian - 8 Gaussian* case, while Table D.1 details results for all 2D synthetic cases. This significant reduction in the target conditional vector field variance enables the objective to provide consistent gradients during training of the network, enabling improved convergence towards identifying the correct disentangled latent dynamics.

**Conditioning Mismatch Kernel**: The conditioning mismatch kernel $\alpha(w)$ plays a pivotal role in the objective introduced in Eq. (13), which we duplicate here in its complete form.

$$\mathcal{L}_{\text{CVFM}}(\theta) = \mathbb{E}_{t,q(z,w),p_t(x|z)p_t(y|w)} \left[ \alpha(w) \| v_\theta(x,y,t) - u_t(x|z) \|^2 \right] \tag{D.2}$$

The selected form of this kernel dictates the degree of continuity expected *a priori* in the observed joint vector fields $u_t(x, y)$ across $y \in \mathbb{R}^M$, introducing an inductive bias in the solution across this joint space, even if we only ever expect to evaluate the vector field in a conditional sense. In this work, we select $\alpha(w) = \exp(-w/2\sigma_y^2)$ with observation $w = y_1 - y_0$, where the degree of continuity in $u_t$ across the conditioning variable can be tailored through an appropriate selection of $\sigma_y$.

Previously, in Table 1, we observed significantly improved performance with the introduction of this additional rebalancing – which naturally raises questions as to its contribution in isolation. Figure D.3 demonstrating that even by itself in the absence ot minibatch OT, the introduction of $\alpha(w)$ is able to reliably equal or improve upon Wasserstein-2 target distribution error in comparison with COT-FM and CVFM. Figure D.4 similarly shows an evaluation across $\eta$ values (the weighting

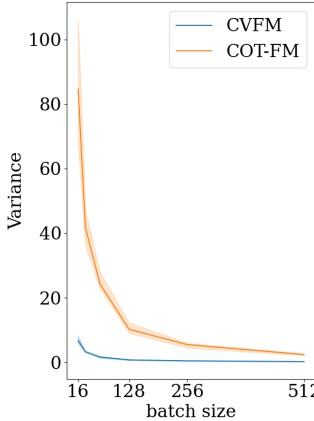

Figure D.2. Objective variance across batch sizes in the *8 Gaussian-8 Gaussian* case.

Table D.1. Variance of the conditional objective across the synthetic datasets considered swept across varying batch sizes. Variance values were computed over 5 seeds with a conditioning transport weight of $\eta = 10$. Values are reported as $\mu \pm \sigma$.

|  | Batch Size | 8 Gaussian-8 Gaussian | 8 Gaussian-Moons | Moons-Moons |
|---|---|---|---|---|
| CVFM | 16 | 6.465±0.561 | 7.847±1.405 | 0.699±0.157 |
|  | 32 | 2.896±0.364 | 3.618±0.142 | 0.179±0.030 |
|  | 64 | 1.515±0.200 | 1.648±0.268 | 0.044±0.007 |
|  | 128 | 0.678±0.0517 | 0.946±0.099 | 0.014±0.002 |
|  | 256 | 0.379±0.086 | 0.493±0.057 | 0.005±0.0005 |
|  | 512 | 0.175±0.032 | 0.216±0.043 | 0.002±0.0004 |
| COT-FM | 16 | 77.376±12.805 | 17.780±2.657 | 3.243±0.356 |
|  | 32 | 42.891±4.948 | 7.482±0.857 | 1.561±0.154 |
|  | 64 | 21.197±0.802 | 4.468±0.621 | 0.739±0.043 |
|  | 128 | 9.672±2.178 | 1.791±0.266 | 0.369±0.049 |
|  | 256 | 5.383±0.588 | 1.057±0.037 | 0.191±0.026 |
|  | 512 | 2.718±0.475 | 0.563±0.045 | 0.094±0.012 |

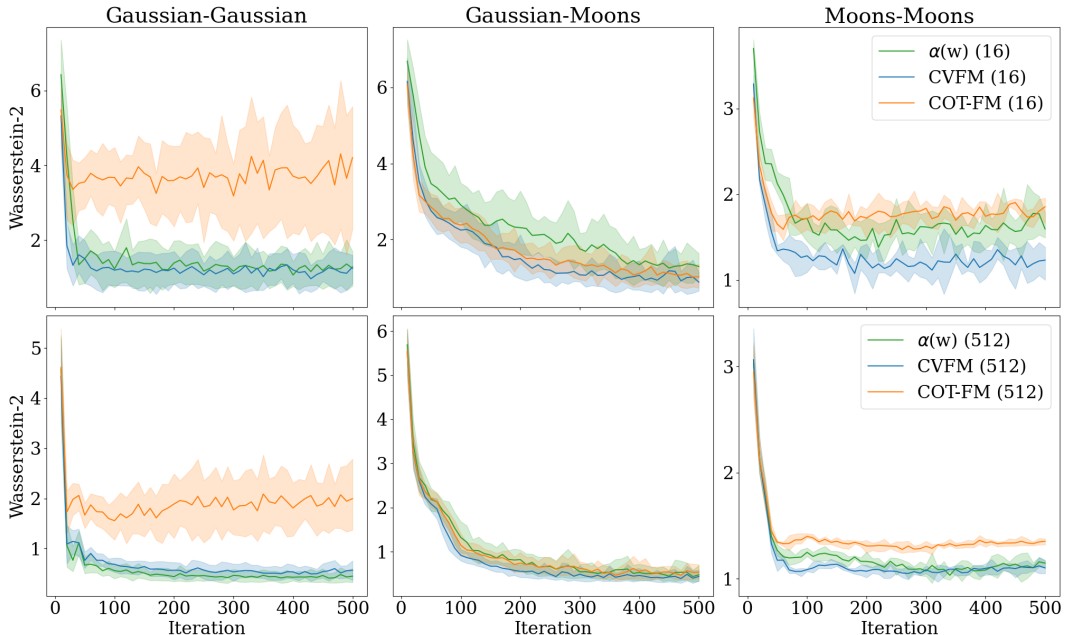

Figure D.3. Demonstration of the effectiveness of performing an expectation of the objective with solely $\alpha(w)$ in comparison to the COT-FM and CVFM approaches. The use of $\alpha(w)$ in isolation is better able to disentangle associated conditioning variables in almost all cases than the conditional Wasserstein distance introduced in Eq. (15).

in the optimal transport ground cost), highlighting the stability it introduces across all test cases relative to minibatch OT solves with Eq. (15). In the *8 Gaussian - 8 Gaussian* case, $\alpha(w)$ drastically outperforms COT-FM, with comparable performance in the other mappings, albeit without providing approximate OT within the conditioned vector fields. Due to this limitation, one might view $\alpha(w)$ in isolation as a reliable extension to conditional CFM.

## D.2 MNIST-FASHIONMNIST DOMAIN TRANSFER

In this section, we further interrogate the results presented in Figures 2 and D.2, and their implications in higher dimensional distributional mappings. The results previously presented

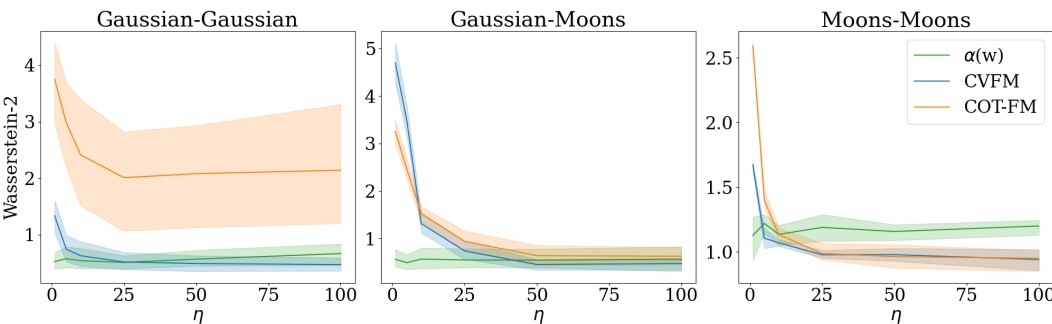

Figure D.4. $\alpha(w)$ more reliably reduces Wasserstein-2 error to the target distribution in comparison with COT-FM across values of $\eta$.

Table D.2. Comparison of conditional mean (FID) and unconditional FID scores (FID-All) for 10,000 samples alongside conditional LPIPS scores.

| Model | FID-All ($\downarrow$) | FID ($\downarrow$) | LPIPS ($\downarrow$) |
|---|---|---|---|
| CVSFM ($\eta = 1000$) | 11.668 | 32.915 | 0.159 |
| COT-SFM ($\eta = 1000$) | 14.965 | 36.456 | 0.161 |
| CVSFM ($\eta = 10$) | 23.554 | 48.326 | 0.175 |
| COT-SFM ($\eta = 10$) | 15.751 | 94.690 | 0.232 |

in Figure 3 illustrated the convergence characteristics of CVFM and COT-FM measured using conditional image metrics. The improved convergence with increasing $\eta$ of COT-FM only further reinforces the value of ground cost scaling across $y$. Unfortunately, the optimal value of $\eta$ is problem dependent, and is particularly challenging to identify a priori. The conditional scaling kernel ameliorates these difficulties. As shown in Figure D.2, the kernel significantly reduces the objective variance in comparison with conditional OT, facilitating stable convergence upon more accurate conditional OT mappings. This behavior was first observed in 2D examples in Figures 2 and D.1 and such characteristics extend to the high-dimensional setting. This increased stability facilitates a greater tolerance on $\eta$ values, ameliorating potential difficulties during hyperparaemter optimization. In Figure D.5, we observe this stability through the inspection of randomly generated samples from mapping the first class of MNIST to the first class of FashionMNIST. While in an unconditional sense, COT-SFM with $\eta = 10$ is able to appropriately transfer between MNIST and FashionMNIST, its consistency in mapping to the correct conditional distribution is lost without elevated penalization in transport across $y$. In comparison, CVSFM is able to consistently map to t-shirts/tops in the first class even with orders of magnitude difference in $\eta$. Similarly, in Figure 3, the convergence behavior for CVFM degrades very little with large changes in $\eta$.

Figure 3 displayed the convergence characteristics of conditional FID and LPIPS scores, conditionally evaluated for each class of $p_1(x|y)$, only serving to reinforce the prior discussion. In comparison, in evaluating FID scores for $p_1(x) = \int p_1(x|y)p(y)dy$, distinctions in the performance between CVSFM and COT-SFM are removed. Figure D.6 highlights the equivalence in unconditional performance in this case study across 10,000 images. This discrepancy in conditional to unconditional FID scores highlights limitations of minibatch sampling from the conditional OT coupling $\pi_\eta((x_0, y_0), (x_1, y_1))$. Even with elevated weighting on transport in the conditioning variable, minibatch conditional OT provides a poorer approximation.

### D.3 MATERIAL DYNAMICS

The mean absolute error metrics displayed in Table 2 provide point estimates of the performance of our proposed method, providing evidence that we are capable of reliably disentangling the latent processing dynamics of material microstructures across the conditional processing space, given unpaired samples of process conditions and material state observed at discrete times. For further interrogation, we present the estimated error distributions in Figure D.7 across all 2,000

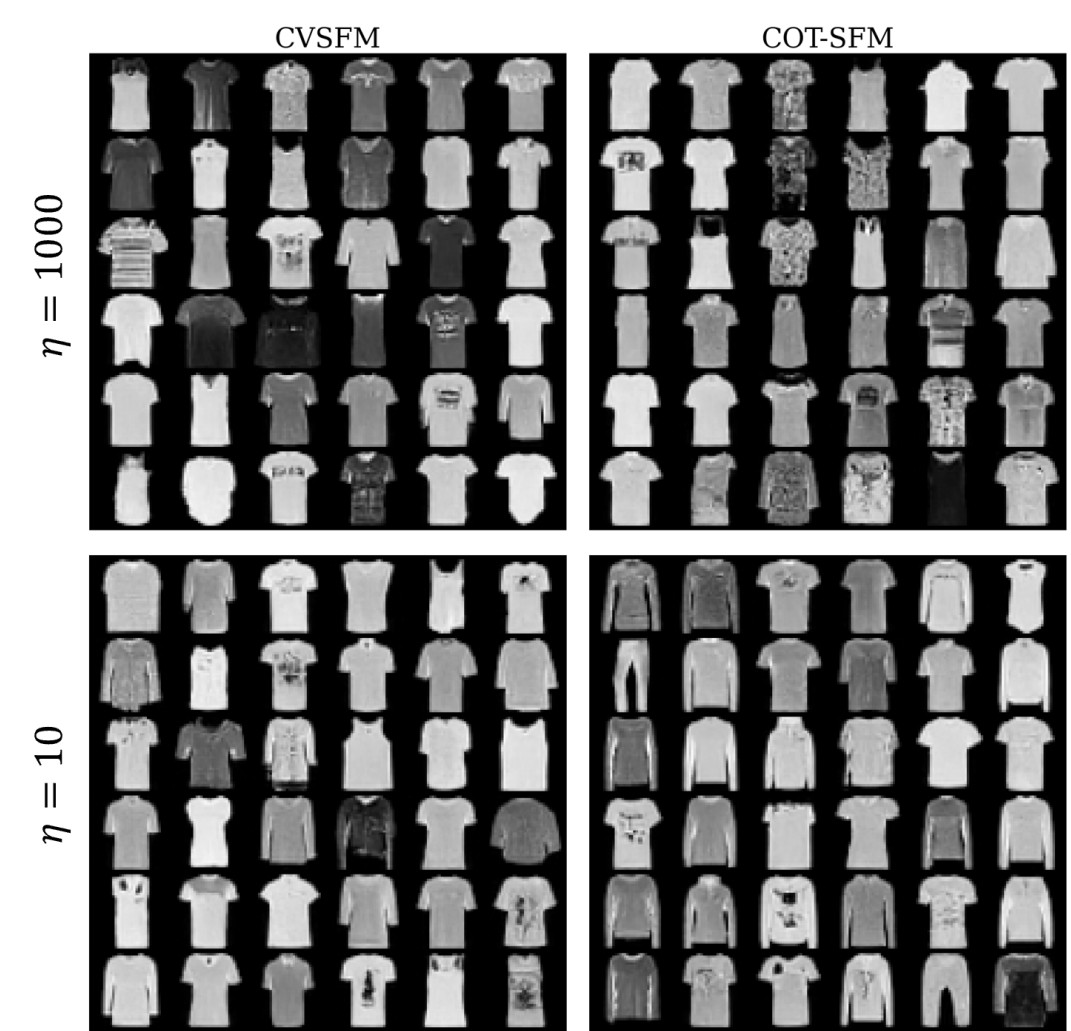

Figure D.5. Comparison of 32 randomly generated images corresponding to the first class of FashionMNIST with CVSFM and COT-SFM for $\eta = 10$ and $\eta = 1000$.

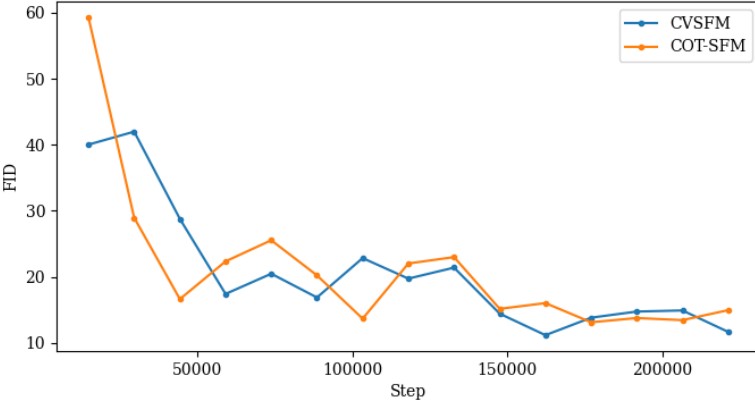

Figure D.6. Unconditional FID scores evaluated across 10,000 samples $x \sim p_1(x)$ for COT-SFM and CVSFM ($\eta = 1000$) during training.

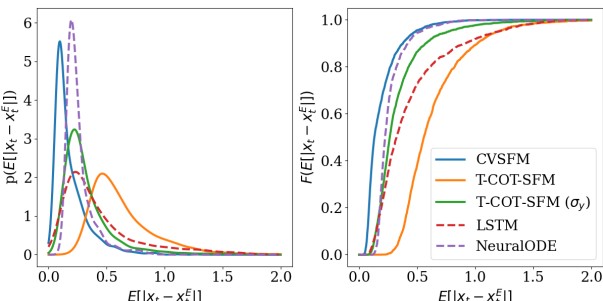

Figure D.7. (Left) probability density function, and (right) cumulative distribution function of CVFM in comparison with evaluated conventional approaches requiring complete trajectory information.

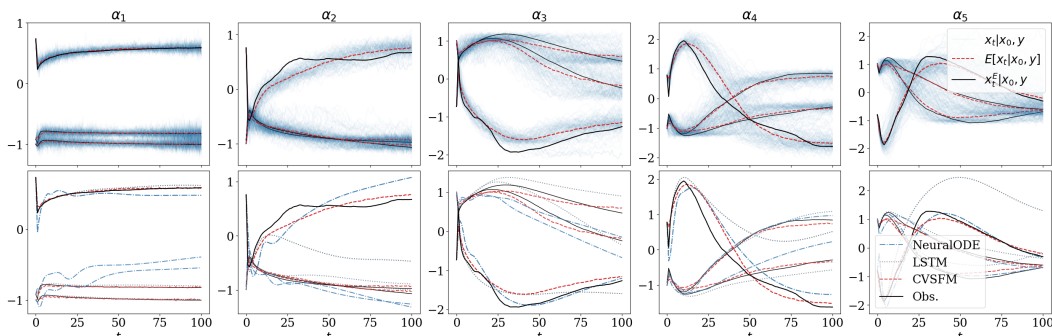

Figure D.8. Collection of three randomly sampled trajectories from the test set in PC space displaying (top) 128 samples in blue from CVSFM, with the expected value in red, and (bottom) deteriministic predictions of the LSTM and Neural ODE shown against the expected value of CVSFM.

material samples in the test set. In congruence with the results presented earlier, approximating the conditional microstructure dynamics through a Neural ODE (Chen et al., 2018) nearly matches the performance of the most optimal CVSFM-Exact variant, exhibiting similar tails and a mean shift – a surprising fact given the stochastic nature of only viewing observations sampled from $q(w, z)$. Figure D.8 compares the dynamics predicted by the three models on several selected members of the test dataset. Mirroring the distribution of errors in Figure D.7, the CVSFM model slightly outperforms the other two while also providing uncertainty estimates. The panels in Figure D.8 display the dynamics projected onto individual principal component subspaces, $\alpha_i$.

# E APPENDIX: EXPERIMENTAL DETAILS

## E.1 ALGORITHM

In this section, we present the general algorithm for Conditional Variable Flow Matching given $q(z, w)$, $p_t(x|z)$, $p_t(y|w)$ in Algorithm 1, and $u_t(x|z)$, and the Schödinger bridge extension with flow and score matching given $\nabla_x \log p_t(x|z)$ in Algorithm 2.

## E.2 STATIC OPTIMAL TRANSPORT

Static exact and entropic regularized optimal transport couplings were solved for in minibatches during training through the Python Optimal Transport (POT) package (Flamary et al., 2021) (https://pythonot.github.io/). As similarly reported in (Tong et al., 2023a), we noticed improved performance in the low-dimensional toy cases with the Sinkhorn algorithm (Cuturi, 2013), which degraded in higher dimensions (e.g., material dynamics, domain transfer). The use of

---

**Algorithm 1** Conditional Variable Flow Matching

---

**Require:** Source and target conditional distributions $q_0(z,w)$ and $q_1(z,w)$, noise $\sigma_x$, $\sigma_y$, and network $v_\theta$.

  **while** Training **do**
    $(x_0, y_0), (x_1, y_1) \sim q(z,w)$
    $t \sim \mathcal{U}(0,1)$
    $\pi \leftarrow \mathrm{OT}((x_0, y_0), (x_1, y_1))$         ▷ Interchangeable with Sinkhorn algorithm Cuturi (2013).
    $(x_0, y_0), (x_1, y_1) \sim \pi(z,w)$
    $\alpha(w) \leftarrow \exp(-(y_0 - y_1)/2\sigma_y^2)$
    $p_t(x|z) \leftarrow \mathcal{N}(x; tx_1 + (1-t)x_0, \sigma_x^2)$
    $p_t(y|w) \leftarrow \mathcal{N}(y; ty_1 + (1-t)y_0, \sigma_y^2)$
    $x \sim p_t(x|z)$
    $y \sim p_t(y|w)$
    $u_t(x|z) \leftarrow x_1 - x_0$
    $\mathcal{L}_{\mathrm{CVFM}}(\theta) \leftarrow \mathbb{E}_{t,q(z,w),p_t(x,y|z,w)} \alpha(w) \| v_\theta(x,y,t) - u_t(x|z) \|^2$
    $\theta \leftarrow \mathrm{Update}(\theta, \nabla_\theta \mathcal{L}_{\mathrm{CVFM}}(\theta))$
  **end while**
  **return** $v_\theta$

---

**Algorithm 2** Conditional Variable Score and Flow Matching

---

**Require:** Source and target conditional distributions $q_0(z,w)$ and $q_1(z,w)$, noise $\sigma_x$, $\sigma_y$, weighting schedule $\lambda(t)$, drift network $v_\theta$, and score network $s_\theta$.

  **while** Training **do**
    $(x_0, y_0), (x_1, y_1) \sim q(z,w)$
    $t \sim \mathcal{U}(0,1)$
    $\pi \leftarrow \mathrm{Sinkhorn}((x_0, y_0), (x_1, y_1), 2\sigma_x^2)$
    $(x_0, y_0), (x_1, y_1) \sim \pi(z,w)$
    $\alpha(w) \leftarrow \exp(-(y_0 - y_1)/2\sigma_y^2)$
    $p_t(x|z) \leftarrow \mathcal{N}(x; tx_1 + (1-t)x_0, \sigma_x^2 t(1-t))$
    $p_t(y|w) \leftarrow \mathcal{N}(y; ty_1 + (1-t)y_0, \sigma_y^2 t(1-t))$
    $x \sim p_t(x|z)$
    $y \sim p_t(y|w)$
    $u_t(x|z) \leftarrow ((1-2t)/(2t(1-t)))(x - (tx_1 + (1-t)x_0)) + (x_1 - x_0)$
    $\nabla_x \log p_t(x|z) \leftarrow (tx_1 + (1-t)x_0 - x)/(\sigma_x^2 t(1-t))$
    $\mathcal{L}_{\mathrm{CVSFM}}(\theta) \leftarrow \mathbb{E}_{t,q(z,w),p_t(x,y|z,w)} \alpha(w) \left[ \| v_\theta(x,y,t) - u_t(x|z) \|^2 + \lambda(t)^2 \| s_\theta(x,y,t) - \nabla_x \log p_t(x|z) \|^2 \right]$
    $\theta \leftarrow \mathrm{Update}(\theta, \nabla_\theta \mathcal{L}_{\mathrm{CVSFM}}(\theta))$
  **end while**
  **return** $v_\theta, s_\theta$

---

minibatch optimal transport has also been previously shown to regularize the transport plan (Fatras et al., 2021b;a) due to the stochastic nature of the independent batch samplings forming non-optimal couplings, in effect resulting in entropic-regularized OT plans, even with exact OT solves.

### E.3 COMPUTATIONAL RESOURCES

All experiments were performed on a high-performance-computing cluster with CPU nodes of 24 CPUs and GPU nodes with V100 and A100 GPUs. 2D experiments were all performed on 1 V100, domain transfer and material dynamics experiments were performed on 4x V100's or 2x A100's.

### E.4 2D EXPERIMENTAL DETAILS

For all 2D synthetic dataset cases we used networks of four layers with width 128 and GELU activations (Hendrycks & Gimpel, 2023). Optimization was carried out with a constant learning rate of $1e-3$ and ADAM-W (Loshchilov & Hutter, 2019) over 10,000 steps and a batch size of

256, unless otherwise specified. Sampling was performed by integration with with the adaptive step size `dopri5` solver and tolerances $\texttt{atol} = \texttt{rtol} = 1e - 5$. Conditional probability paths $p_t(x|z)$ were defined with $\sigma_x = 0.1$, which was held constant throughout all cases. Values of $\eta$ and $\sigma_y$ for $p_t(y|w)$ were varied between discrete and continuous conditioning cases. In discrete conditioning cases (*8 Gaussian - 8 Gaussian* and *8 Gaussian - Moons*), $\sigma_y = 0.02$ and $\eta = 100$, whereas in the continuous conditioning case (*Moons - Moons*), $\sigma_y = 0.5$ and $\eta = 5$.

The assignment of classes for each of the 2D synthetic datasets was performed in the discrete conditioning cases (e.g., 8 Gaussian - 8 Gaussian and 8 Gaussian - Moons) by maintaining the distinct 8 classes between the source and target densities. In the case of the 8 Gaussian - 8 Gaussian mapping, each of the Gaussians maintained it's own class assignment. For the 8 Gaussian - Moons mapping, the 2 Moons were discretized with assignment provided based on relative angle to the origin of each particular moon. Classes for each moon were broken into 4 groups based on 45º degree segments. The Moons - Moons mapping was established with a 90º degree rotation of the target density about the origin, and continuous conditioning assignment provided by the expression $y = (x_0 - 10)I_{[0,1]} + (1 - I_{[0,1]})(x_0 + 10)$, where $I_{[0,1]}$ denotes the initial binary moon classification.

Empirical values of the Wasserstein-2 distance were evaluated through 2,048 samples simulated through the learned conditional vector fields and computed against an equivalent number of samples from the target distribution. The $W_2$ reported distance differs depending on cases with continuous or discrete conditioning. In the discrete case, we take the mean of the conventional $W_2$ distance across all conditioning classes

$$W_2(\hat{p}_1, q_1) = \mathbb{E}_{y \sim p(y)} \left[ \left( \inf_{\pi \in \Pi(\hat{p}_1, q_1)} \int \|x_i - x_j\|^2 d\pi(x_i, x_j) \right)^{1/2} \right] \tag{E.1}$$

while in the continuous case, we incorporate the conditional ground cost as

$$W_2(\hat{p}_1, q_1) = \left( \inf_{\pi \in \Pi(\hat{p}_1, q_1)} \int \left[ \|x_i - x_j\|^2 + \eta \|y_i - y_j\|^2 \right] d\pi((x_i, y_i), (x_j, y_j)) \right)^{1/2} \tag{E.2}$$

with $\eta = 1e5$. Both distances are computed between samples from the target distribution $q_1$, and samples from $q_0$ simulated forward to $t = 1$ as $\hat{p}_1$.

Similar to Tong et al. (2023b;a) and Shi et al. (2023), we also report the normalized path energy with a similar continuous/discrete split, either as the expected value across all classes in the discrete case, or with the cost as in Eq. (15). For brevity, we only repeat the normalized path energy with this latter formulation

$$\text{NPE}(q_0, \hat{p}_1) = \frac{\left| \mathbb{E}_{q_0(z,w)} \int \|v_\theta(x, y, t)\|^2 dt - W_2^2(q_0, \hat{p}_1) \right|}{W_2^2(q_0, \hat{p}_1)}. \tag{E.3}$$

This quantity provides a normalized measure for the deviation of the path energy learned by the model $v_\theta$ to that which would be optimal, equivalent to the squared Wasserstein-2 distance denoting constant velocity trajectories.

### E.5 MNIST-FASHIONMNIST EXPERIMENTAL DETAILS

The MNIST-FashionMNIST domain transfer experiments utilized a UNet architecture developed by OpenAI (https://github.com/openai/guided-diffusion/tree/main). The network configuration utilized in this work consisted of 64 channels, with channel multiples of [1,2,2,2]. 4 heads of self-attention over 16 and 8 resolution were applied with 2 residual blocks. Optimization was carried out with cosine annealing of the learning rate from $1e - 4$ to $1e - 8$ and ADAM-W Loshchilov & Hutter (2019) with weight decay $1e - 4$ for ranges of 3,750 - 10,000 epochs and batch size of 1024, or equivalently 220,000 - 590,000 steps. Conditional probability paths were constructed with $\sigma_x = 0.1$, $\sigma_y = 1e - 3$, and minibatch conditional OT was performed with $\eta = 10$ and $\eta = 1000$.

Unconditional FID scores were computed over 10,000 samples using *Clean-FID* (`https://github.com/GaParmar/clean-fid`). Conditional FID scores were computed using the same algorithm, however the calculation was restricted to a subset of the total samples with matching conditioning (1,000 samples per class). LPIPS scores were computed using *torchmetrics* (`https://github.com/Lightning-AI/torchmetrics`).

### E.6 MATERIAL DYNAMICS EXPERIMENTAL DETAILS

All material specific cases used networks with five layers of width 256 and GELU activations (Hendrycks & Gimpel, 2023). Skip connections were applied over the middle three layers. The conditioning variable was subject to both self-attention and time cross-attention with an embedding dimensionality of 64 and 8 heads. The same network architecture was used in T-COT-FM, and for both drift and score networks in CVSFM, COT-SFM, and T-COT-SFM. It was duplicated for the NeuralODE benchmark. The LSTM benchmark consisted of four layers with width 512 for approximately equivalent parameterization. Optimization was carried out with cosine annealing of the learning rate from $1e-3$ to $1e-8$ and ADAM-W Loshchilov & Hutter (2019) with weight decay $1e-2$ over 7,800 steps and a batch size of 256. Conditional probability paths were constructed with $\sigma_x = 0.1$, $\sigma_y = 0.01$, and minibatch conditional OT was performed with $\eta = 10$. While, T-COT-FM in it's initial formulation does not include a varying noise schedule across the conditioning variable, we have also included an ablation with this extension, mirroring the same value of $\sigma_y$.

The available dataset was established through spinodal decomposition simulations in MEMPHIS (Dingreville et al., 2020). The dataset contains 10,000 two-phase microstructures of size $256 \times 256$ voxels, each associated with processing parameters $\theta_1$ and $\theta_2$, sampled according to the log-uniform $\log(\theta) \sim \mathcal{U}(\log(0.1), \log(100))$, and $\theta_3 \sim \mathcal{U}(-0.7, 0.7)$. These processing parameters correspond to the mobility parameters of the two constituents $(\theta_1, \theta_2)$, and initial relative concentrations $(\theta_3)$. Mean absolute error values, along with minimum and maximum absolute error reported in Table 2 were evaluated on a test set of 2,000 microstructures, and a training set of 8,000 microstructures. 2-point statistics were computed for all 100 recorded frames, constituting the trajectory of an individual microstructure, and compressed via Principal Component Analysis (PCA). All models were trained on the first 5 PC dimensions.

