# OpenReview forum: "Conditional Variable Flow Matching: Transforming Conditional Densities with Amortized Conditional Optimal Transport"
_ICLR.cc/2025/Conference — ICLR 2025 Conference Withdrawn Submission_

### Official Review · Reviewer_o1P6 · 2024-10-18

**Soundness:** 3
**Presentation:** 3
**Contribution:** 2
**Rating:** 6
**Confidence:** 3

**Summary:**

This paper propose conditional variable flow matching (CVFM), a framework for learning to transform conditional distributions between source and target distributions given unpaired samples (response and covariates are not in correspondence). This is achieved by simultaneously sample $x$ and $y$, and further incorporate $y$ into smoother (neural net) for vector field. To further stabilize and accelerate training, they further use conditional OT & mismatch. In the experiement part, they extensively study the proposed method.

**Strengths:**

Further extend the CGFM (guided flow), by further do sampling on covariates $y$. Therefore, we can deal with the unpaired data.

**Weaknesses:**

Even before the CGFM formally released, the conditional sampling strategy by incorporating $y$ into neural net has been used widely in many fields. This work further extends the CGFM, but may raise some concern for novelty?

**Questions:**

1. In line 364, not sure I understand your claim that CGFM cannot extend to conitnuous conditioning. Since $y$ can be continuous, if we include $y$ into neural net, we can do conditional sampling based on continuous covariate $y$?
2. Out of curiousity: the CGFM can still used for unpaired data by simply do local permutation/ resampling of covariates $y$ during training? Is this strategy not ideal for efficiency? Just a quick idea.

---

> ### Author Response · Authors · 2024-11-22
>
> Thank you for taking the time to read through our paper and for your thoughtful comments and suggestions. First, regarding the unique changes to CVFM; we certainly agreed that incorporating $y$ into a network is rather trivial. However, there are three additional nontrivial changes necessary to extend flow matching to continuous conditional distributions. We believe that this is where the primary innovations lie. As we have discussed in the other reviewer responses, these innovations are the inclusion of flow over both variables, including a conditional wasserstein distance, and a conditional loss reweighting kernel. Importantly, all of these changes are theoretically motivated based on a presented mathematical analysis (Thrm 3.1 and Appendix A). Provably, the optimal transport in the conditioning is necessary for stable flow on these problems. We also demonstrate this empirically through experimentation. For example, in Figure 2 and Table 1, we compare CVFM against several other variants, including an approach where $y$ is simply added to the network with no other framework changes (labeled CFM). We show that the performance of such a na\"{i}ve approach is significantly reduced -- as theoretically expected. The violation of OT over y, and the consequentially degraded performance, is precisely why we cannot directly include our mismatched $(x_0,y_0), (x_1,y_1) \sim q(z,w)$ when $y_0 \neq y_1$ into the network.
>
> Regarding your comments about CGFM; in scientific applications, we are frequently presented with disjoint multi-marginal empirical distributions with $y \in \mathbb{R}^n$ (i.e., with continuous conditioning variable). In this setting, CGFM cannot function as there is no capability to resample because it is statistically impossible to sample $\{(x_0, y_0), (x_1, y_1) \}$ pairs such that $y_0=y_1$ when $y$ is continuous and the mini-batch size is finite. By introducing the conditional ground cost in Eq. (15), we provide support across $y$ in our optimal coupling $\pi_{\eta}(x,y)$ (or $\pi_{\eta}(z,w)$ over empirical marginal samples), enabling us to resample according to this map during training. We also analyze further, and demonstrate that solely using this coupling (incorporated in both COT-FM and T-COT-FM) is insufficient in practice, as highlighted in Table 1 and Figure 2 for the 2D case studies, Figure 3 in our domain transfer case study, and Table 2 and Figure 4 in our microstructure dynamics example. Regarding discrete problems, without both continuous flow matching frameworks cannot match CGFM's performance. We believe that this decrease in performance is owed to the hyperparameter $\eta$, which in COT-FM/T-COT-FM must be carefully selected and is mapping specific. Whereas by introducing the kernel $\alpha(w)$, we provide greater tolerances on $\eta$, facilitating greatly improved performance as emphasized in the previously mentioned Figures and Tables, alongside additional discussions in Appendices D1 \& D2. These appendices highlight the significantly reduced objective variance of CVFM in relation to COT-FM in 2D case studies. In our domain transfer case studies, we demonstrate that COT-FM provides equivalent FID scores in our target FashionMNIST distribution (Appendix D.2 - Figure D.5 \& Table D.2), although underperforms in class-specific FID scores (Figure 3). In other words, it often generates examples from the wrong class when drawing conditional samples.
>
> Overall, our proposed method presents a straightforward methodology for extending FM to the conditional setting. In Appendix E.1, we present ODE/SDE-based algorithms incorporating our proposed changes, which provide no additional overhead over CFM with minibatch OT Tong et al. (2023) \& Pooladian et al. (2023).

---

> > ### Comment · Reviewer_o1P6 · 2024-11-23
> >
> > Thanks to the author for addressing my questions and suggestions. At least based on the numerical results provided in the paper, the CVFM improves performance in terms of flow stability and sampling quality.
> >
> > However, I’m still confused about the scientific motivations for "unpaired data." Even for the example provided in the paper, we can still formalize it as a regression problem, and hence it is "paired" to some degree. Whether it’s truly "paired" or not, the data will show by itself. Moreover, even though the resampling is not possible statistically (as argued in the rebuttal), we can still do permutation (e.g. with extra training iterations)?
> >
> > Anyway, I have currently raised the rating to borderline accept. Either accepting or rejecting is reasonable to me, and I will let the AC make the decision.

---

> > > ### Author Response · Authors · 2024-11-25
> > >
> > > Thank you again for the feedback on this work. In another reviewer response, we begin to highlight some of the motivations behind this work, although to provide additional clarity, this method was developed while working with materials manufacturing data we unfortunately do not have permission to disclose (motivating our last case study). In the process design and optimization of complex manufacturing routes (1000+ steps), pieces being processed with varied process conditions are sampled for inspection at intermediate steps throughout the workflow. At each of these junctures, samples are then sent for destructive examination of the material samples' internal state/microstructure, enabling downstream interrogation of linkages between these states and quality or yield metrics.
> > >
> > > From this illustrative example, it becomes clear that CVFM becomes directly applicable when we are aiming to model stochastic dynamics across some conditional space (e.g., process parameters) without access to complete trajectory information. There are undoubtedly similar examples across the sciences as sampling limitations and destructive examination are widespread.
> > > Returning to our last case study, you are absolutely correct to point out that it could in fact be posed as a regression problem. This is precisely what we benchmark against in our simple NeuralODE and LSTM cases, where these two architectures are provided complete trajectory information which the alternate methods are not. By being able to simulate this problem in both settings, we can directly contrast the ability of CVFM to disentangle conditional dynamical processes, even in comparison to directly regressing upon their latent trajectories.
> > >
> > > Lastly, one of the core complications is that we are specifically targeting samples where $y \in \mathbb{R}^n$, but is also highly informative of the underlying dynamics. Permuting across our empirical samples $(x_0, y_0), (x_1, y_1) \sim q(z,w)$ without addressing the conditional mismatch in this setting would provide equivalent performance as our investigated ‘CFM’ method in Table 1 and Figure 2 of our first case study.
> > >
> > > Once again, thank you for taking the time out to review our work, and we appreciate all of the feedback provided.

---

### Official Review · Reviewer_CNND · 2024-10-23

**Soundness:** 4
**Presentation:** 3
**Contribution:** 3
**Rating:** 6
**Confidence:** 4

**Summary:**

The authors introduce conditional variable Flow Matching (CVFM), which allows for the transformation of conditional distributions across continuous conditioning variables with unpaired samples. The paper demonstrates the application of CVFM across various domains, such as image-to-image domain transfer and modeling the temporal evolution of materials' internal structures during manufacturing processes, showing improved performance and convergence characteristics over existing methods.

**Strengths:**

* The paper is generally well-written and easy to follow, with a few exceptions noted below.

* The experiments are well chosen, ranging from controlled synthetic scenarios to real-world material-related challenges.

* There are informative additional experiments and ablations in the appendix which demonstrate the effectiveness and superior performance of the proposed method.

**Weaknesses:**

* There are typos in some equations that need correction.

* The descriptions of some experiments are unclear, particularly how discrete and continuous variables are used.

* The discussion on how the proposed model differs from or outperforms the $[SF]^2$M (Tong et al, 2023) approach is vague and requires clarification. Please refer to the questions below for specifics

**Questions:**

* Equation 6, typo in the subscript. It should be $\mathbb{P}_0 = \nu_0$ rather than $ \nu_1$, and $\text{argmin}$ should be used instead of $\text{min}$. Please review Equation 6 carefully.

* What’s the main difference between the proposed method and  $[SF]^2$M (Tong et al, 2023)?. Section 3.1 of  (Tong et al, 2023) suggests $[SF]^2M$ model can handle general conditioning information $z$, where pair $(x_0, x_1)$ is a special case. It appears the proposed method by the authors may be a special case of  $[SF]^2$M? Can the authors clarify the difference? Or why is the proposed method better than  $[SF]^2M$ ?

* In the experiments. I got confused about discrete and continuous conditioning used.  In Figure 2 (8 Gaussian-Moon),  what are the conditional variables $y_0$ and $y_1$ associated with $x_0$ and $x_1$ ? Intuitively, the authors might align 8 labels to $x_0$, but what are the conditional variables for $x_1$? Besides, is one-hot encoding applied to the discrete conditional variables to differentiate it from continuous conditioning variables? The same question for the material dynamics example

---

> ### Author Response · Authors · 2024-11-22
>
> Thank you for the diligent review, and for catching these oversights on our part regarding the manuscript. To address the questions raised:
>
> * Thank you again for catching these oversights in Eq. (6). They have been suitably addressed in an updated version of the paper.
>
> * The main theoretical difference between CVFM and $[SF]^2M$ is that CVFM can handle samples with unpaired values of their conditioning variables. This extension to enable the amortized approximation of the conditional vector field $u_t(x|y)$ represents our core contribution, demonstrating how to accept both unpaired samples $(x_0,y_0), (x_1,y_1) \sim q(z,w)$ without restriction that $y_0 = y_1$ on a sample basis. Data with this structure is not uncommon in industrial manufacturing applications. Importantly, the transition to handling fully unpaired samples (i.e., unpaired in both the primary variable, $x$, and the conditioning variables, $y$) is not as simple as Tong et al.'s paper might suggest. In our paper, we try our best to show this both theoretically and practically. Theoretically, our Thrm. 3.1 (proof in Appendix A) notes a subtle difference: flow matching on conditional distributions is only stable without transport in the conditioning variable. In practice, this observation motivates the primary contributions of CVFM: a conditional OT distance and a conditional reweighting kernel.
>
> 	In the paper, we compare against CFM (a flow matching model where the conditioning is directly embedded) and repeatedly see decreased performance. In fact, for many problems, CFM does not converge (as theoretically predicted). We also show that both the conditional distance and the kernel are necessary to stabilize training.
>
> 	Its worth noting that Tong et al.'s $[SF]^2M$ framework is readily compatible with our extensions. In fact, we rely heavily on the sde + flow paradigm they introduce for improved results on both the image to image domain transfer case study and the microstructure dynamics case study.
>
> 	Its also worth noting that while in the setting of discrete conditioning, one can readily sample the empirical marginal densities $\{z \sim q(z|y_i)\}^{m}_{i=0}$, this is infeasible for many scientific applications, where continuous conditioning (e.g. process parameters in manufacturing settings) is frequently encountered. This limitation of both CFM and consequently $[SF]^2M$ can be seen in Figure 2 \& Table 1, where we specifically highlight CFM. Due to the mismatch across the conditioning variable, the framework consistently struggles to correctly disentangle the desired dynamics. This is a result of Theorem 3.1, demonstrating COT is a requirement for satisfying the continuity equation. Despite CFM with minibatch OT Tong et al. (2023) and $[SF]^2M$ Tong et al. (2023) incorporating minibatch resampling in $x$, both frameworks only serve to sample $z \sim \pi(z)$ according to the optimal coupling across $x$, neglecting $y$ information and resulting in cross-conditioning pairings.
>
> * Thank you for pointing this out, we should have made these details for our 2D examples much clearer in our appendices. We have since addressed this and provided clarified experimental details in Appendix Section E.4. More specifically, The the 8 Gaussian - 2 Moon 2D case study discretizes the 2 Moons distributions about the center point of each moon by grouping points into equivalent conditioning every 45 degrees.
>
> * In the discrete case the method would be broadly applicable with either one-hot encoding or ordered integer values, although we emphasize that the primary motivation for this method lies in applicability to scientific applications with associated continuous conditioning. In our last case study, the conditioning for the microstructure evolution consists of a continuous range of mobility and relative concentration values for the 2-phase constituent microstructures considered. These are considered as processing parameters within our spinoidal decomposition simulations, discussed in Appendix B.1.

---

> > ### Comment · Reviewer_CNND · 2024-11-25
> >
> > Thank you for the clarifications provided in your response, which adequately address my concerns. I appreciate the efforts made to enhance the manuscript in other aspects as well. Based on these improvements, I am inclined to recommend acceptance of the paper and have increased my confidence score accordingly.
> >
> > Additionally, I suggest using a different color to highlight any changes made in the revision. This would enable reviewers to quickly identify and track the modifications, especially since the original draft was deleted and pdfdiff cannot be used.

---

> > > ### Author Response · Authors · 2024-11-25
> > >
> > > Thank you for the increased vote of confidence in the manuscript, and we sincerely apologize for the oversight in not highlighting updates to the .pdf. We will make sure to provide additional clarity to changes/updates in any future revisions. Thank you again for taking the time to review our work.

---

### Official Review · Reviewer_hYuT · 2024-11-03

**Soundness:** 2
**Presentation:** 2
**Contribution:** 1
**Rating:** 3
**Confidence:** 4

**Summary:**

This paper studies flow matching in the setting of conditional simulation. The authors study conditional optimal transport as a tool for designing couplings in this setting, and empirically verify their proposed method on several 2D datasets, an image-to-image problem, and a material dynamics problem.

**Strengths:**

- The paper is generally clear, well-written throughout
- The problem studied is important and likely of relevance to the broader community
- Generally the derivations in the paper are sound

**Weaknesses:**

### Novelty / Related Work
- A major weakness of this paper seems to be novelty. There are several recent works which study conditional OT for flow matching. It was not clear to me what is new in this submission that did not appear in these prior works. In particular, both of these papers use the weighted cost in Equation 15 to build minibatch COT couplings, followed by fitting a flow matching model using these couplings. There are many more similarities as well, e.g., both papers identify the $q(y_0) = q(y_1)$ condition appearing in Theorem 3.1 of this submission. These both appeared in March/April 2024 (~5 months before the ICLR deadline), so should probably not be counted as concurrent works.
     - [Dynamic Conditional Optimal Transport through Simulation-Free Flows, NeurIPS 2024](https://arxiv.org/abs/2404.04240)
     - [Conditional Wasserstein Distances with Applications in Bayesian OT Flow Matching, arXiv](https://arxiv.org/abs/2403.18705)

- Prop. 3.1 in the submission is known, see e.g. [Carlier 2008](https://arxiv.org/abs/0810.4153), and this work should be cited for the modified loss in Equation 15 and the corresponding convergence in Prop 3.1. Moreover, the two papers above use this result and the corresponding loss function in Equation 15 to derive COT methods for flow matching.

- In Lines 90-01, the authors write "leveraging a novel conditional Wasserstein metric..." and similar things in the abstract.
     - The authors never explicitly describe what this metric is (or e.g. prove that it is a metric).
     - Conditional Wasserstein metrics have already been developed and are very similar to the notions discussed in the submission. Please see the two linked papers and this [PhD Thesis](https://ricerca.sns.it/retrieve/e3aacdfd-ef40-4c98-e053-3705fe0acb7e/Gigli_Nicola.pdf), Chapter 4, for previous work in this direction.

### Experiments
- There are no baselines for the MNIST-FMNIST image transfer experiment. The baselines in Table 2 are quite weak (only an LSTM and a neural ODE). Table 1 only compares against ablations of the proposed method.

**Questions:**

Please see weaknesses

---

> ### Author Response · Authors · 2024-11-22
>
> We appreciate you taking the time to review our submission in such detail, and thank you for passing along several additional references. We've taken your feedback and made several updates to our submission, including:
>
> * Included citations to Carlier et al. (2008) with respect to Prop. 3.1.
> * Performed ablations with respect to the implementation of Kerrigan et al. (2024) in both our 2D case studies and our microstructure dynamics case study (T-COT-FM).
> * Adjusted portions of the abstract and introduction for additional clarity, highlighting the salient differences between our methods and those listed in your review (improved performance, decreased sensitivity to hyperparameters, and theoretical differences in the proofs).
> * We also clarified our wording to refer to usage of a conditional Wasserstein distance, not a metric.
>
> CVFM certainly addresses a comparable problem to the two papers you have identified. However, we believe that it makes several important contributions to a very nascent conversation on this topic (also discussed in our response to reviewer one). We argue that learning flows on conditional distributions requires careful practical incorporation of conditioning into both the architecture and the training framework - involving three changes. Like Chemseddine et al. and Kerrigan et al., we motivate these changes by presenting a theoretical proof (Appendix A). We argue that OT coupling between the marginal distributions over the conditioning variables is mandatory. Chemseddine et al. arrive at a similar conclusion through a different argument, however, Kerrigan et al., do not; they argue that any couplings is acceptable but OT is a good choice. With this knowledge, CVFM focuses on subtle, but important, practical changes to the flow matching training framework to promote this theoretical requirement: (1) conditional OT mini-batch coupling via a combined distance over the conditioning and primary variables (as you note: present in both Chemseddine et al. and Kerrigan et al.), and (2) kernel reweighting the loss based on distance in the conditioning space.
>
> Both additions are necessary to achieve good performance. Additionally, in our experiments, we observed that the kernel facilitates model training and less sensitive to $\eta$, observed in our updated Table 1 \& Table 2. The conditional cost function denoted in Eq. (15) only approximately provides COT, representing a necessary relaxation over $y$, but degrading performance relative to CGFM (which samples $\{z \sim q(z|y_i)\}^{m}_{i=0}$ during training for each conditional empirical density). COT-FM (i.e., without the kernel) performed better in comparison with CFM, also improving with increasing batch size. However, CVFM outperforms both -- in some examples, by a significant margin. The same behavior is also observable in Table 1, Fig. 3, and Table 2 (the materials dynamics problem -- the primary foreseen application of CVFM). In our updated document for the material dynamics case study, we also compare against T-COT-FM (Kerrigan et al. (2024)). To facilitate a fair comparison, we significantly extend their proposed implementation: adding variable noise scheduling in the conditioning variable and including score matching Tong et al. (2023). With these changes, the model is similar to the COT-FM discussed above but still underperforms. As an aside, we observed that it was quite important to separately tune conditional probability paths via $\sigma$ and $\sigma_y$ over the main and conditioning variables, significantly improving performance in Kerrigan's model and our own, Fig. 4.
>
> Finally, we also demonstrate that adding the kernel decreases the sensitivity of conditional flow matching frameworks to choice of the noise schedule (Appendix D -- D.2) and the conditional distance hyperparameter, $\eta$. This can be seen in both Table 1 \& Figure 2 (2D toy problems) and Figure 3 (image to image domain transfer). In practice, this makes the model's hyperparameters much easier to optimize. We believe that the primary reason for this benefit is that the kernel reduces the variance of the training objective. In Figure D.2 \& Table D.1 in the Appendix, we see that the kernel decreases the variance by almost an order of magnitude for small batch sizes. Figure D.4 additionally provides an ablation across $\eta$ values from 0-100 against the target 2-Wasserstein distance. The figure illustrates that CVFM demonstrates significantly less sensitive to this specification as result of the weighting kernel. For some problems, similar results can be achieved using COT-FM with careful hyperparameter tuning. But, performance is never exceeded.
>
> In total, the added kernel is not a trivial addition; it significantly improves performance in practice. We make several changes to the paper to (1) highlight these two recent works, (2) identify similarities, and (3) emphasize the differences described above and the important contributions of our work.

---

> > ### Comment · Reviewer_hYuT · 2024-11-26
> >
> > I thank the authors for their response. However, I do not believe the updated version of the submission appropriately addresses the differences between this work and the previous works I mentioned in my review. I hope the other reviewers will take the time to read these related papers as well. Moreover, the authors do not address my concerns regarding baselines in the MNIST-FMNIST experiment or Table 2.
> >
> > According to your response (and my own understanding of these works), the main novelty in your approach which does not appear in these prior works is the kernel $\alpha(w)$ appearing in Equation 13, as well as some minor practical changes (e.g., the noise scheduling you describe).
> >
> > While this does seem to provide additional practical benefits (as demonstrated in your experiments), the paper is written in such a manner to suggest that the entire framework of flow matching with continuous $y$ using optimal transport techniques is new. This is not the case given these prior works. Overall, I strongly recommend the authors re-frame their proposed approach (which, to be clear, I think has potential merit given the empirical results) to more clearly and explicitly discuss what precisely is new in the submission and what has already appeared in the literature.
> >
> > I provide here several specific examples which appear in the latest version of the submission which I do not think fairly position this work in relation to the existing literature.
> > - Abstract:
> >      - "While flow-based models can impressively predict the temporal evolution of probability distributions representing possible
> > outcomes of a specific process, existing frameworks cannot satisfactorily account for the impact of conditioning variables on these dynamics. Amongst several limitations, existing methods require training data with paired conditions and are developed for discrete conditioning variables."
> >    - The authors seem to be suggesting here that their work is the first to handle unpaired data and continuous conditioning variables. This is not the case -- the papers I link explicitly do this.
> > - Line 326
> >    - "Instead, we moderate this requirement in a manner similar to concurrent proposals by Kerrigan et al. (2024) and Chemseddine et al. (2024) in the form of the proposed continuous non-negative cost function ..."
> >    - Your proposal is not "similar" to these works -- it is precisely the same.
> > - Line 253
> >    - "In the following section, we generalize the FM objective in Eq. (3) to matching the flow between arbitrary conditional distributions, p0(x|y) to p1(x|y), provided unpaired observations with distinct conditioning variables..."
> >    - Again, these constructions appear explicitly in both of the papers I linked.

---

> > > ### Author Response · Authors · 2024-11-27
> > >
> > > Thank you once more for taking the time to diligently review our work. We appreciate the continued feedback, which we believe has, and will only serve to further strengthen this paper in the future. Unfortunately, due to the time limitations for submission of a revised document, we no longer believe it would be possible to suitably address your comments in the time allotted, and as such will withdraw our submission from further consideration.
> > >
> > > Thank you again once more for your time and valuable feedback.

---

### Official Review · Reviewer_47wU · 2024-11-04

**Soundness:** 3
**Presentation:** 3
**Contribution:** 2
**Rating:** 3
**Confidence:** 4

**Summary:**

the authors present CONDITIONAL VARIABLE FLOW MATCHING (CVFM), a model for generative modeling within the flow matching framework, where a neural network learns the implicitly defined vector field between conditional distributions. They do this by pairing samples across source and target distributions by penalising cross-condition assignment and learning condition dependent flows.

**Strengths:**

I am drawn to the idea of extending the flow matching idea to other settings, such as conditional distributions. CVFM does so in a simple, straightforward way, which consists of smart sampling of source and target pairs, and conditioning of the neural network. The authors also extend the theory of FM and prove the holding of the continuity to the conditional case.

The paper is generally well written and the authors have put emphasis on readability and having their manuscript be easy to follow. Their empirical validations, particularly the results regarding material trajectory inference, are convincing and demonstrate the validity of their approach.

**Weaknesses:**

The approach presented in this paper is relatively incremental. As mentioned, CVFM consists of two improvements beyond standard FM, the first is using a conditional-attenuated distance for OT based source and target assignment, and the second is the conditioning the model's neural network via the input. While the first is cool, it is not clear to me if it holds enough merit on its own, as It's hard to claim the neural network component is all that novel, as training FM by conditioning the input (i.e. learning f(x,y,t), instead of f(x,t)) is a straightforward extension. The authors should make clear what sets their approach apart from the current state of the field.

The impact of the condition coefficient eta eq 15 needs to be better studied. As mentioned, this is distance is a key conceptual contribution of this, and demands careful ablation. In the supplement, I see values ranging for 5 to 1e5 (lines 1534 and 1549). How sensetive is the algorithm to choice of eta? In the discrete case, is this even that relevant? Can you not just sample equally from each condition and only pair within?

CVFM is only relevant when both source and target distributions are influenced by conditions. While its ubiquitous to have structured data be the target distribution (i.e. images separated by class), the source is usually chosen to be an easy to sample from noise distribution, such as the uniform or standard Gaussian. It is not immediately clear to me from this paper what are the instances when there is a need to transform structured source into structured target. This issue is somewhat palpable based on the results presented in this work, which are either simulated datasets (8 Gaussians & Moons) or a conditional transfer between MNIST and FashionMNIST images. The authors should make further attempts to demonstrate their work on real-world applications besides a single material trajectory inference example. This would serve to both further corroborate their work while also convincing the community of its relevance.

Figure 2 could be made clearer. In the 8 Gaussian and Moons datasets, what are the different conditions? This information particularly pertinent for concept behind this work. I understand the trajectories are coloured by condition but it's not clear what they are.

**Questions:**

The authors make note that CVFM amortises Flow Matching over conditionals. This implies an important advantage: instead of learning separate flows for each condition, it learns a single model that can handle all conditions by sharing information across them. To properly validate this claim, it would be valuable to include a comparison against a baseline where you train separate flow matching models for each condition individually. This experiment would directly test whether CVFM's unified approach actually performs better than having dedicated models for each condition.

Can the authors please share code? The main contribution of this work is algorithmic, access to code would greatly improve my ability to judge the merits of this manuscript.

Is the pseudo-code in algorithms 1 & 2 correct? As written, the values for pt(x|w) and pt(y|w) are computed before the OT assignment, but if thats so, the x and y value in lines 1476 & 1504 do not align with the regressed flow.

---

> ### Author Response · Authors · 2024-11-22
>
> Thank you for taking the time to thoroughly review of our work and catching a few oversights. We have since updated the mistake noted in the ordering of the algorithms in the Appendix (i.e., shifting sampling of $y \sim p_t(y|w)$ and $y \sim p_t(x|z)$ following OT resampling), and made adjustments to further clarify points outlined below.
>
> Learning mappings across conditional distributions requires careful practical incorporation of conditioning into both the architecture and the training framework -- involving three changes motivated by a proof for satisfying the continuity eq. (Thrm. 3.1 and Appendix A). This theorem is subtly -- but importantly -- different from those in prior works; it identifies that OT coupling between the marginal distributions over the conditioning variables at $t=0$ and $t=1$ is mandatory. Given this, CVFM introduces three changes: (1) conditioning as an input in the neural network (trivial), (2) conditional OT mini-batch coupling, and (3) kernel reweighting the loss based on distance in the conditioning space, made to mitigate the approximation of minibatch OT and the relaxation in the COT cost Eq. (15).
>
> (2) and (3) are necessary for good performance. For example, Fig. 2 shows a comparison of the performance between CFM (trivial inclusion of $y$), COT-FM (CVFM without the kernel -- approximately similar to a concurrently developed approach by Chemseddine et al. (2024)), and CVFM. CFM does not converge (Thrm 3.1). COT-FM improves, particularly with larger batch sizes. CVFM outperforms both -- in some examples, by a significant margin. The same behavior is also observable in Table 1 (2D problems), Fig. 3 (image to image domain transfer), and Fig. 4 / Table 2 (materials dynamics problem -- the primary foreseen application of CVFM). In our updated document, we add an additional comparison in the materials dynamics problem. We compare against a concurrently developed model by Kerrigan et al. (2024): T-COT-SFM. To facilitate a fair comparison, we significantly extend their proposed implementation: adding variable noise scheduling in the conditioning variable, and including score matching Tong et al. (2023) -- still the model underperforms. The added kernel is not a trivial addition; it significantly improves performance in practice.
>
> As you've highlighted, the hyperparameter $\eta$ significantly impacts model performance. However, this is primarily true for the T-/COT-FM ablation. The added kernel significantly reduces sensitivity (Appendix D.2). This can be seen in both Table 1 \& Figure 2 \& Figure, allowing for easier optimization of the model's hyperparameters by reducing objective variance by nearly an order of magnitude for small batch sizes (Appendix Figure D.2). To clarify a small confusion: in all studies we set $\eta < 1000$, only elevating to $\eta = 1e5$ for computing reported conditional 2-Wasserstein distances. Figure D.4 additionally provides an ablation across $\eta$ values from 0-100, further illustrating this reduced sensitivity to $\eta$. In some problems, COT-FM can achieve similar results carefully tuning $\eta$, but, performance is never exceeded.
>
> The development of this method was mainly targeted with scientific applications in mind -- frequently characterized by continuous conditioning and messy, unpaired data. It would be impossible to sufficiently sample each condition to apply methods such as CGFM (representing precisely what you are suggesting, requiring the ability to sample $\{z \sim q(z|y_i)\}^{m}_{i=0}$ from each of the $m$ discrete classes). This limitation is also why CGFM cannot be evaluated on our 2 Moons - 2 Moons case study (Table 1 \& Figure 2). We have added additional clarifying statements to address this misunderstanding in Section 4. Directly comparing CVFM  against learning separate flows for each condition, is in fact the benchmark CGFM model in Table 1, naturally providing the best performance where applicable, although it is incapable of extending to our applications of interest. Even with CVFM only provided misaligned samples, it most closely approaches this lower-limit performance.
>
> Regarding Figure 2, we entirely agree, and it is an oversight on our part. We have aimed to clarify this point by including additional information in Appendix E.4 for the interested reader on how the specific 2D case studies were created, particularly emphasizing the conditioning present in each. We have also made our code available to interrogate these specific case studies further.
>
> We've have put up the code on an anonymized repo to maintain anonymity. An example notebook can be found here: https://anonymous.4open.science/r/conditional-variable-flow-matching-332E/examples/cvfm_tutorial.ipynb. We'd also like to highlight that the role of $\alpha(w)$ is critical from a practical standpoint, as evidenced in our first two case studies, while demonstrating significant performance improvements over solely the COT cost in Eq. (6).

---

> > ### Comment · Reviewer_47wU · 2024-11-22
> > **Response to rebuttal**
> >
> > Dear authors,
> >
> > I really appreciate your response and for taking the time to revise the manuscript and add additional experiments. As I've stated in my original review, I am partial to the ideas presented in this paper and I do like the manuscript. However, after reading the response, I cannot raise my score beyond the original 3.
> >
> > While some of my points were addressed, namely typos and figure clarity, my main concerns still stand. I find this work incremental, and in my opinion a straightforward modification of the OT-based mini-batch assignment within the Flow Matching training process does not constitute enough novelty for this venue.
> >
> > This is not to say I have something against simple ideas, on the contrary. But they need to work well and solve a problem other approaches cannot. In my experience with Flow Matching, simply adding the condition to the neural network works reasonably well with generating specific samples.
> >
> > I believe this work has potential, but so far I'm not convinced of its merits.

---

> > > ### Author Response · Authors · 2024-11-22
> > >
> > > Thanks again for your prompt feedback, although we believe there may still be some miscommunication. We were quite surprised by your anecdote. In our experience, when the conditioning values in the samples are mismatched ($y_0$ != $y_1$) in a minibatch during training, it does not converge to a reasonable mapping. We should clarify that this method is specifically meant for the situation in which you have a collection of points $\{ (x^j_0, y^j_0), (x^j_1, y^j_1) \}_{j=1}^n$ from arbitrary $p_0$ to arbitrary $p_1$ with entirely mismatched values $y$. To the best of our knowledge, apart from the ablations highlighted, there are no other methods of achieving this, despite its simplicity in implementation.
> > >
> > > Thanks again for your time and suggestions.

---

> > > > ### Comment · Reviewer_47wU · 2024-11-22
> > > > **Response to authors**
> > > >
> > > > Dear Authors,
> > > >
> > > > Allow me to clarify. My anecdote was referring to when just the target data has conditioning values, not for transporting specific instances of source to specific instances of target. Indeed, that is different scenario than the one your manuscripts deals with, but my point still stands.
> > > >
> > > > I believe I understood well your manuscripts contributions.

---

### Note · Authors · 2024-11-28

I have read and agree with the venue's withdrawal policy on behalf of myself and my co-authors.